# Differential conformational expansion of NUP98-HOXA9 oncoprotein from nanosized assemblies to macrophases

Hao Ruan[1,2,6], Rodrigo F. Dillenburg [3,6], Elnaz Hosseini[4], Sina Wittmann [4,5], Martin Girard [3,5] ✉ & Edward A. Lemke [1,4] ✉

Transcription factors (TFs) play a central role in gene regulation by binding to specific DNA sequences and orchestrating the transcriptional machinery. A majority of eukaryotic TFs have a block copolymer architecture, with at least one block being a folded DNA interaction domain, and another block being highly enriched in intrinsic disorder. In this study, we focus on NUP98-HOXA9 (NHA9), a chimeric TF implicated in leukemogenesis. By integrating experiments and simulations, we examine the structural dynamics of NHA9's FG domain across assembly states. We find that the FG domain has different conformational compactness in the monomeric, oligomeric, and densely packed condensate state. Notably, the oligomeric state exhibits micelle-like organization with non-fixed stoichiometry, with the DNA-binding domain exposed at the periphery. These findings offer molecular insight into the phase behaviour of NHA9 and highlight dynamic conformational transitions of intrinsically disordered regions during molecular assembly, with implications for understanding transcriptional regulation in cancer.

Gene expression regulation at the level of transcription relies on the binding of transcription factors (TFs) to gene regulatory regions. TFs recognise and bind to specific DNA sequences within promoter or enhancer elements, subsequently recruiting coactivators, chromatin remodelers, and modifiers. These components collaborate to assemble the preinitiation complex, which includes RNA polymerase II (RNA Pol II), thereby initiating transcription[1]. The activity of TFs is tightly regulated through various mechanisms that ultimately shape gene expression and influence a broad range of cellular processes[2]. Eukaryotic TFs consist of multiple functional domains, among which, arguably, intrinsically disordered regions remain the least characterised, despite their essential roles[3]. Notably, over 80% of eukaryotic TFs contain at least one intrinsically disordered region, which are often substantially longer, frequently exceeding 250 amino acid residues—than those found in other eukaryotic proteins[4–6].

Recent studies have revealed that TFs, RNA Pol II, mediator, and other transcription-associated components can self-organise into droplet-like structures frequently referred to as transcriptional condensates[7–11]. These condensates have been implicated in enhancing the spatial and temporal regulation of transcription and thus gene expression[12–19]. In many cases, multivalent intermolecular interactions mediated by intrinsically disordered regions within TFs can serve as a driving force for condensate formation. Intrinsically disordered proteins (IDPs, or proteins rich in intrinsically disordered regions, which we here collectively refer to as IDPs) also promote conformational flexibility, facilitating interactions with diverse partners[8,20,21]. The abundance of IDPs is even more pronounced in chimeric TFs. Those are often disease-associated and formed by the fusion of the DNA-binding domain (DBD) of one gene with one or more transcriptional regulatory domains from another[22].

[1]Biocenter, Johannes Gutenberg University Mainz, Mainz, Germany. [2]Institute of Molecular Biology postdoctoral program, Mainz, Germany. [3]Max-Planck Institute for Polymer Research, Mainz, Germany. [4]Institute of Molecular Biology (IMB gGmbH), Mainz, Germany. [5]Institute for quantitative and computational biosciences, Mainz, Germany. [6]These authors contributed equally: Hao Ruan, Rodrigo F. Dillenburg. ✉e-mail: martin.girard@mpip-mainz.mpg.de; edlemke@uni-mainz.de

In this study, we investigate the chimeric TF NUP98-HOXA9 (NHA9), one of the most common NUP98 fusion proteins resulting from the *t(7;11)(p15;p15)* chromosomal translocation, which is associated with acute myeloid leukaemia, myelodysplastic syndrome, and chronic myeloid leukaemia[23–25]. Notably, ectopic expression of NHA9 has been shown to induce leukaemic transformation in mouse models[26]. The N-terminal region of NHA9, derived from a component of the nuclear pore complex, contains two IDPs enriched in phenylalanine-glycine (FG) repeats, separated by a folded GLEBS domain responsible for binding to the RAE1 protein (Fig. 1a)[27]. Recent studies have demonstrated that NHA9 undergoes phase separation to form condensates, and hundreds of chromatin-associated puncta have been observed in the nuclei of cells[28–30]. Furthermore, NHA9 condensates have been shown to alter higher-order genome organisation, thereby modulating gene transcription during leukemogenesis[29,31]. Increasing evidence indicates that many phase-separating proteins can form heterogeneous clusters even at concentrations below the phase separation saturation threshold[32–35]. Thus, the questions emerge if such nanoclusters might also exist for oncogenic fusion proteins like NHA9, how potentially diverse assembly states affect NHA9 conformation, and what functional implications this might have. While for IDPs an actual structure-function relationship as for folded proteins, is not easily rationalised, the conformational ensemble is of major relevance for the IDP to execute its function. Experimentally addressing this challenge is difficult due to the large molecular size, inherent heterogeneity and flexibility of long IDPs. Similarly, tackling this

computationally is challenging due to the substantial computing power and prolonged simulation times required.

Herein, we conduct a detailed characterisation of NHA9 structure and dynamics using a combination of single-molecule fluorescence resonance energy transfer (smFRET) spectroscopy, fluorescence lifetime imaging of fluorescence resonance energy transfer (FLIM-FRET), coarse-grained simulations and phase separation assays. These approaches are employed to investigate the self-association behaviour of the FG domain of NHA9 across different assembly states. Our findings indicate that the FG domain of NHA9 undergoes continuous expansion while transitioning from a dilute monomeric to an oligomeric state, and eventually to a densely packed phase. Notably, the oligomeric state of NHA9 exhibits micelle-like structural features, with the hydrophilic DBD positioned near the surface–resembling the behaviour of an amphiphilic diblock copolymer. Although their architecture is non-random and capable of binding DNA, their size depends on the concentration of NHA9, consistent with non-core-shell architectures. Taken together, the continuous transition from a dilute solution to micelle-like formation and ultimately to macrophase separation indicates that NHA9 behaves as a weak amphiphile.

## Results
### IDP of NHA9 is dynamic and highly compacted in dilute solution
Structural and disorder predictions of NHA9 suggest that the FG domains remain disordered, while the HOXA9 DNA-binding domain retains a folded and binding competent conformation (Fig. 1b).

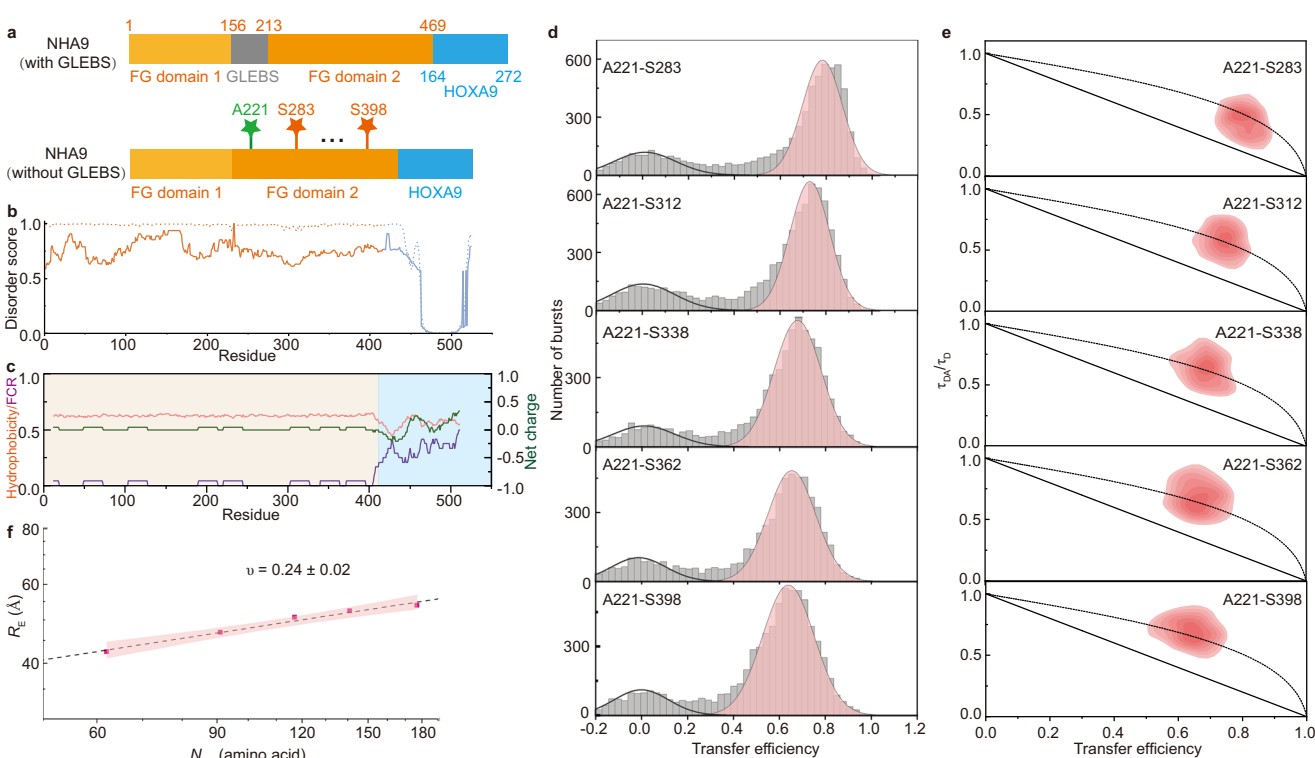

**Fig. 1 | NHA9 FG domain is dynamic and highly compact in the dilute state.**
**a** Schematic representation of the NHA9 construct with and without GLEBS domain. The constructs include FG nucleoporin domain 1 (yellow), the GLEBS domain (grey), FG nucleoporin domain 2 (orange), and the HOXA9 domain (blue). Residues used for fluorescence labelling are indicated. **b** Disorder prediction as a function of residue number, calculated using two different predictors: NetSurfP-3.0[88] (dashed line) and DEPICTER[89] (solid line). The FG domain is highlighted in yellow and the C-terminal HOXA9 is in blue. **c** Sequence property profile of NHA9 calculated using the HPS-Urry force filed[57]. Hydrophobicity (λ parameter) is shown in red, the fraction of charged residues (FCR) in purple and net charges in green.

Yellow regions denote the hydrophobic IDP block, while blue highlights the hydrophilic HOXA9 block. **d** Single-molecule transfer efficiency histograms of NHA9 in the dilute state. The small peak at $\langle E \rangle \approx 0$ corresponds to donor-only labelled molecules. **e** Two-dimensional histogram of relative donor fluorescence lifetime ($\tau_{DA}/\tau_D$) versus transfer efficiency, $\langle E \rangle$, calculated from individual bursts. The dashed line represents the dynamic line, based on a Gaussian chain polymer model. See Methods section for details. **f** Mean inter-residue distances extracted from smFRET data and scaling law fitting. $\nu$ is presented as the fitted value ± the standard error of the estimate. The red shaded area indicates the 95% confidence interval of the fit. Source data are provided as a Source Data file.

Structurally, NHA9 resembles a diblock copolymer, in which the disordered FG domain functions as a hydrophobic block, while the structured DBD serves as a hydrophilic block (Fig. 1c).

To investigate this domain's conformational behaviour and dynamics, we turned to smFRET[36–38]. A key advantage of smFRET is its ability to monitor specific subregions within structurally heterogeneous samples and to work at low concentrations where aggregation phenomena are less likely to occur. We purified NHA9 from *E. coli* inclusion bodies under denaturing conditions to perform our measurements. However, the structured GLEBS domain has previously been shown to facilitate molecular ageing/aggregation[39,40]. It has been shown that deletion of the GLEBS domain (that sits between the FG domains 1 and 2, Fig. 1a) does not impair NHA9-mediated transformation of primary hematopoietic stem and progenitor cells. Therefore, we purified NHA9 lacking the GLEBS domain for in vitro experiments (Supplementary Fig. 1a)[29]. We note that even without the GLEBS domain, the protein shows a strong tendency to aggregate even at low concentrations and was thus kept in a denaturing buffer until rapid dilution into physiological conditions prior to any of the following experiments. To probe conformational behaviour of the IDP domain of NHA9, we focused on the FG domain and created different FRET donor-acceptor pairs and labelled the FG domain 2 site-specifically by incorporating the unnatural amino acid p-acetylphenylalanine (AcF) at position 221, which was conjugated to Alexa488-hydroxylamine (donor dye). We selected five acceptor labelling sites located in inter-FG spacers approximately every 30 residues (Supplementary Table 2). In choosing these sites, we aimed to avoid phenylalanine, glycine, charged residues, and highly hydrophobic residues. By introducing a cysteine at various positions, we generated five FG domain variants of NHA9 for labelling with Alexa594-maleimide (acceptor dye; Fig. 1a). The labelled proteins were analysed at the single molecule level during free diffusion through the confocal volume of the microscope to determine the mean transfer efficiency $\langle E \rangle$ of the FG domain. In the resulting smFRET histograms, a peak near $\langle E \rangle \approx 0$ corresponds to donor-only species, while a second population represents molecules containing functional donor-acceptor pairs. All five constructs exhibited Gaussian-shaped FRET histogram distributions with histogram widths consistent with the behaviour of a dynamic, rapidly fluctuating ensemble of IDPs (Fig. 1d).

To quantify the dimensions of the FG domain of NHA9 from smFRET data, we converted the mean transfer efficiencies into the root-mean-squared end-to-end distances ($R_E$) between the dye pairs. This conversion was based on intrachain distance distributions modelled as a Gaussian chain, an established approximation for unfolded proteins[41,42]. To assess chain compaction independently of sequence length, we calculated the apparent Flory scaling exponent, $\nu$, using the scaling relation $R_E \propto N^\nu$, which relates the $R_E$ to chain length ($N$)[43]. The FG domain of NHA9 exhibited a compact conformation in dilute solution, with a scaling exponent $\nu = 0.24 \pm 0.02$ (Fig. 1f), in line with the hydrophobic nature of its FG motifs leading to collapse of the protein due to strong intramolecular interactions[44]. To validate these measurements, we also performed lifetime-based analysis, which yielded $R_E$ values in line with the intensity-based analysis (Supplementary Fig. 3). To probe rapid conformational dynamics of the FG domain, we utilised relative fluorescence lifetime measurements to detect distance fluctuations between fluorophores on timescales between the fluorescence lifetime (nanoseconds) and the interphoton waiting time (microseconds). Analysis of donor fluorescence lifetimes revealed systematic deviations from the static FRET line in normalised donor lifetime versus transfer efficiency plots (Fig. 1e), consistent with a broad distribution of distances characteristic of a rapidly fluctuating IDP ensemble.

## NHA9 forms nanoclusters in subsaturated solutions with reduced compactness

Recent studies have identified that for a few IDPs, stable clusters can exist both below and above the critical concentrations at which macroscopic phase separation occurs[32,33,45–48]. To test for such an effect for NHA9, we first characterised the phase behaviour of NHA9 in vitro using turbidity assays. Turbidity increased sharply above around 2 µM, indicating the onset of macroscopic condensate formation (Supplementary Fig. 5a). Therefore, we focused our nanocluster measurements on concentrations below 2 µM. Due to the inherent possibility for the existence of heterogeneous nanoclusters, accurately characterising the nanocluster size and conformation can be challenging. Fluorescence correlation spectroscopy (FCS) enables the determination of molecular size by analysing diffusion dynamics. We employed repetitive FCS measurements with short 10-second sampling times to reduce interference from large aggregates on the signal. Specifically, LD655-labelled NHA9 at a concentration of 10 nM was used as a fluorescent probe in the presence of varying concentrations of unlabelled NHA9. Autocorrelation analysis of repeated FCS measurements yielded a distribution of average diffusion coefficients $\langle D \rangle$ by single-component fitting. NHA9 exhibited a bimodal size distribution, consisting of monomers (and small oligomers) and larger multimolecular nanoclusters (Fig. 2a). As the concentration of unlabelled NHA9 increased, the larger nanoclusters with heterogeneous size distributions progressively emerged.

To probe conformational changes associated with nanocluster formation, we conducted smFRET experiments using FRET-labelled NHA9 to examine transfer efficiency histograms of monomeric versus nanocluster states. Solutions containing 50 pM of FRET-labelled NHA9$_{A221-S362}$ were doped with increasing concentrations of unlabelled NHA9. With increasing concentration of unlabelled NHA9, we observed a second peak appearing at a lower transfer efficiency, indicating the FG domain of NHA9 adopts a more expanded conformation in nanoclusters compared to monomers (Fig. 2b). The relative abundance of this low-efficiency peak increased at the expense of the higher-efficiency peak, consistent with nanocluster growth observed via FCS (Fig. 2c). Notably, the mean transfer efficiency $\langle E \rangle$ of the lower peak decreased from $0.40 \pm 0.12$ at 100 nM to $0.34 \pm 0.15$ at 2 µM (Fig. 2b, Supplementary Fig. 5b), suggesting a progressive reduction in compactness of the FG domain with increasing nanocluster size.

We extended these observations to four additional FG-domain variants, each with an increasing distance between the labelling sites, all of which exhibited a second, lower-efficiency peak at a concentration of 1 µM (Fig. 2d). As we also observed acceptor quenching during nanocluster formation (see Supplementary Fig. 4), intensity-based analysis was not used to extract distance. Instead, end-to-end distances $R_E$ were obtained from lifetime-based fitting of only the donor signal using the Gaussian chain model according to Eq. 11 (see "Methods" for details). The FG domain displayed a larger apparent Flory scaling exponent ($\nu = 0.33 \pm 0.08$) in nanoclusters at a concentration of 1 µM compared to the monomeric state, indicating reduced compactness (Fig. 2e). Additionally, all five labelling variants deviated from a simple linear relationship in a 2D histogram of lifetimes versus intensity-based transfer efficiency, indicating that the FG domain also remains disordered and dynamic within the nanocluster assembly state (Supplementary Fig. 5c). In summary, combining FCS and smFRET analyses revealed a concentration-dependent increase in nanocluster size and abundance. Within the nanoclusters, the FG domain of NHA9 exhibits reduced compactness, which appears to correlate with nanocluster size.

## Impact of DNA binding on NHA9 nanocluster formation

To determine whether DNA binding affects nanocluster formation, we first confirmed the DNA-binding ability of purified NHA9 using a gel

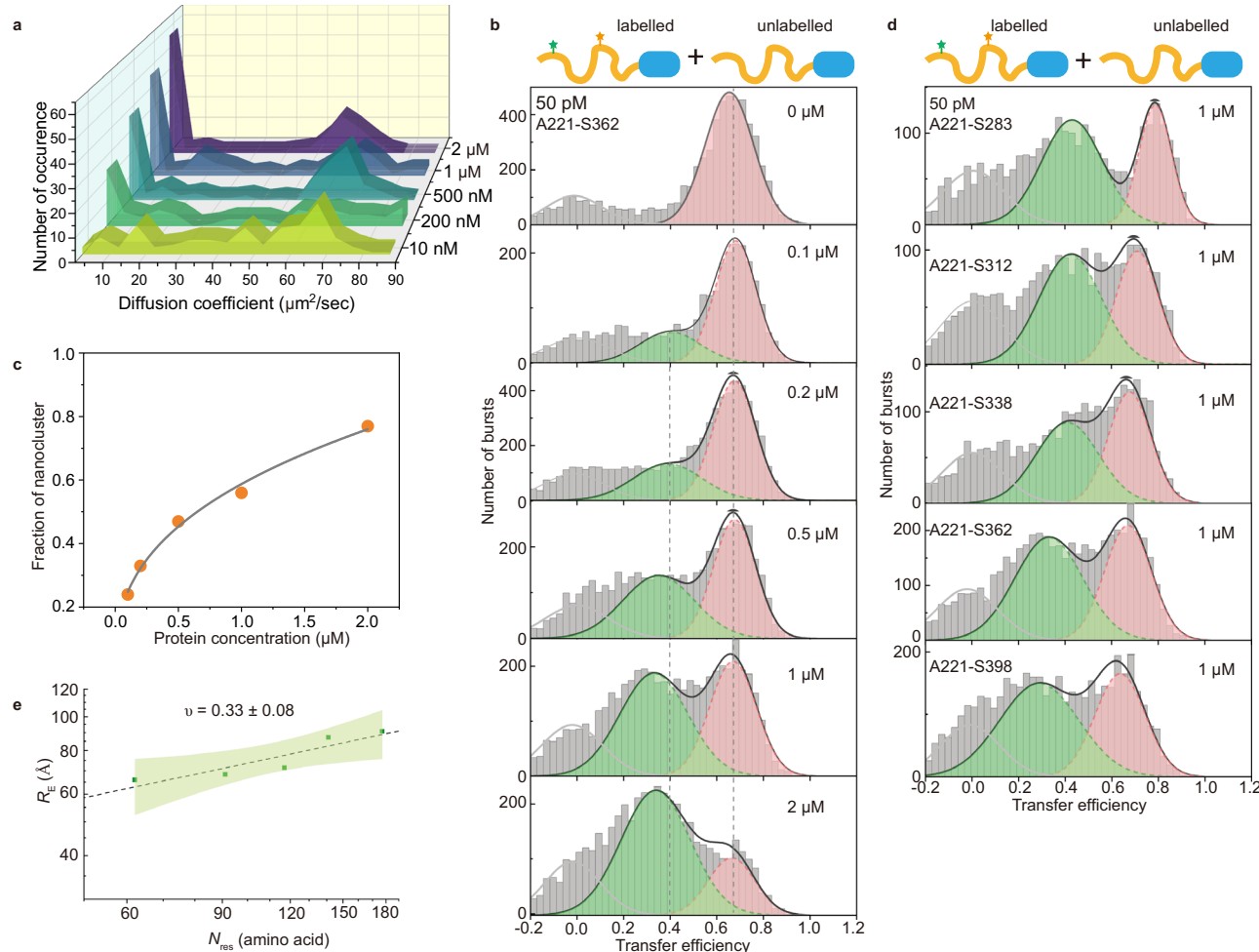

**Fig. 2 | NHA9 forms nanocluster in subsaturated solutions and shows less compactness within nanocluster. a** Concentration-dependent nanocluster formation of NHA9 measured by FCS. Autocorrelation curves were obtained from repeated (120) measurements with short sampling time (10 s), using NHA9$_{A221C}$ labelled with LD655 as fluorescent probe in trace amounts (10 nM) with excess unlabelled NHA9. The distribution of the average diffusion coefficient at different concentrations of total protein was plotted. **b** Single-molecule transfer efficiency histograms of NHA9$_{A221-S362}$ at increasing concentrations of unlabelled protein.

The peak near $\langle E \rangle \approx 0$ corresponds to donor-only species. **c** The fraction of nanocluster extracted from Fig. 2b as a function of overall protein concentration. **d** Single-molecule transfer efficiency histograms of five FRET labelled NHA9 variants at 1 μM. **e** $R_E$ extracted from lifetime-based fitting and scaling law of the NHA9 FG domain within nanoclusters. $v$ is presented as the fitted value ± the standard error of the estimate. The green shaded area indicates the 95% confidence interval of the fit. Source data are provided as a Source Data file.

mobility shift assay (Supplementary Fig. 2a). This assay employed a 20-base-pair double-stranded DNA oligonucleotide containing the HOXA9 binding site[30]. The results confirmed that NHA9 interacts specifically with the DNA fragment containing the HOXA9 consensus binding site, yielding an apparent $K_D$ of 0.3 μM. As a control, we also performed a gel mobility shift assay with a sequence-shuffled DNA that had an identical nucleotide composition but lacked the HOXA9 consensus site to assess whether this effect was sequence-specific. As a result, NHA9 exhibited weaker binding with a shuffled DNA fragment (Supplementary Fig. 2b). Furthermore, both circular dichroism spectroscopy and nano-differential scanning fluorimetry measurements demonstrated that the DBD of NHA9 purified under denaturing conditions rapidly refolds upon transfer into native buffer (Supplementary Fig. 1c, d). We then investigated the effect of DNA binding on nanocluster formation using FCS. Repeated FCS measurements with short sampling intervals revealed a marked reduction in the population of larger nanoclusters upon addition of the DNA fragment (Fig. 3a). At lower protein concentration (1 μM NHA9), this effect was less pronounced (Supplementary Fig. 6b), possibly due to a compensatory mechanism: DNA binding can solubilize NHA9 molecule and removing them from the nanocluster, which is offset by the added

mass of those DNA molecules that stay bound to the nanocluster. In line with this, FRAP experiments also showed a slight mobility increase of NHA9 in the presence of DNA (Supplementary Fig. 6a). To further evaluate this phenomenon, we analysed smFRET histograms of NHA9 at 1 μM and 2 μM NHA9, each in the presence of 5 and 20-fold excess DNA. As DNA concentration increased, the fraction of nanoclusters decreased, while the monomer fraction increased, consistent with FCS observations (Fig. 3b, Supplementary Fig. 6d). Additionally, we detected a mild but consistent increase of $\langle E \rangle$ from $0.34 \pm 0.15$ to $0.37 \pm 0.12$ and $0.38 \pm 0.1$ at DNA concentrations of 10 μM and 40 μM, respectively, suggesting that DNA binding slightly reduced the size of NHA9 nanoclusters. As a control, the $\langle E \rangle$ values of monomers remained constant after adding DNA (Supplementary Fig. 6c). As another control, we also tested the shuffled DNA fragment. While this control fragment can also inhibit nanocluster formation, its effect was substantially weaker than that of the specific HOXA9-binding DNA, as evident from the maintenance of a much larger fraction of the low $\langle E \rangle$ population (Supplementary Fig. 6d). These results indicate that specific interactions at least partially mediate the reduction in nanocluster size between NHA9 and its target DNA sequence. However, nonspecific nucleic acid interactions may also play a role. This result may seem

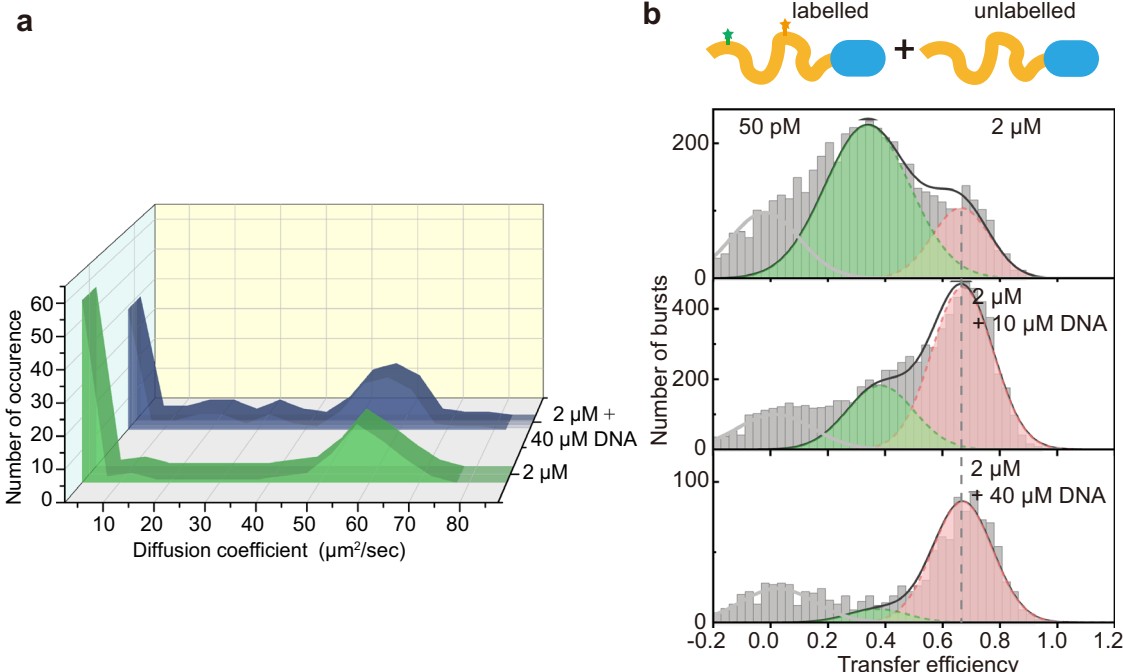

**Fig. 3 | DNA influences NHA9 nanocluster formation. a** Nanocluster formation of NHA9 at 2 μM with or without the presence of a DNA fragment measured by FCS. Autocorrelation curves were obtained from repeated (120) measurements with short sampling time (10 s), using NHA9$_{A221C}$ labelled with LD655 as fluorescent probe in trace amounts (10 nM) with excess unlabelled NHA9. **b** Single-molecule transfer efficiency histograms of NHA9$_{A221-S362}$ at 50 pM labelled protein and 2 μM of unlabelled protein in the presence of different concentrations of DNA fragment. Source data are provided as a Source Data file.

counterintuitive, as DNA was previously reported to enhance the condensation of TFs[49,50]. As we are using short DNA oligonucleotides to which only around one NHA9 molecule can bind, we reason that the DNA is unable to promote nanocluster formation.

To further investigate the driving force of NHA9 nanocluster formation, we performed smFRET measurements of 1 μM NHA9 protein in buffer supplemented with 5% or 10% 1,6-hexanediol. In the native buffer, half of the NHA9 population was in heterogeneous nanoclusters. This fraction was reduced to 25% in a buffer containing 5% 1,6-hexanediol and further reduced to an undetectable level in a buffer containing 10% 1,6-hexanediol (Supplementary Fig. 7). These results confirm that hydrophobic interactions primarily drive NHA9 nanocluster formation. Upon binding to DNA, the NHA9–DNA complex exhibited greater solubility compared with NHA9 alone. Furthermore, electrostatic repulsion between DNA molecules within a nanocluster may further enhance the solubilization of NHA9.

## NHA9 undergoes significant conformational expansion in macroscopic condensates

Building on our observations of intramolecular conformational changes within nanoclusters, we investigated whether a shift in conformational distribution also occurs during macroscopic phase separation. Accurate measurement of the mean transfer efficiency ⟨E⟩ in protein droplets is challenging due to their inherently high background fluorescence, which can obscure the signal. To address this, we turned to fluorescence lifetime imaging of FRET (FLIM-FRET), which enables the extraction of FRET efficiency based solely on the donor's fluorescence lifetime, thereby eliminating the need for direct acceptor excitation and detection. Given the strong background signal in droplets, which makes FRET intensity correction more challenging, we employed ensemble FLIM-FRET measurements instead of single-molecule level measurements. The FLIM-FRET method avoids common pitfalls of intensity-based FRET methods, such as spectral spillover, and simplifies experimental design by reducing the need for extensive controls and normalisation[40,51]. To further minimise

background signal, we utilised the AF594 and LD655 dye pair to label NHA9. FRET efficiency was determined by measuring the reduction in donor fluorescence lifetime upon energy transfer to the acceptor, allowing us to estimate average spatial distances between fluorophores. Following sample preparation (see Methods), we doped a 10 μM solution of unlabelled NHA9 with 1 nM of dual-labelled NHA9 and performed FLIM-FRET imaging of droplets forming on coverslips after triggering phase separation by rapid dilution from the denatured NHA9 stock solution into native buffer (Fig. 4a).

Analysis of donor fluorescence lifetime decay curves across five NHA9 variants revealed a slower decay with increasing chain length ($N_{res}$) between the FRET labels, indicating reduced FRET efficiency in longer chains (Fig. 4b). We also converted individual fluorescence decay curves into a phasor plot, which enables the graphical representation of FLIM data. In the phasor plot, each point corresponds to the fluorescence decay profile of a single sample. Phasor analysis revealed some degree of heterogeneity among samples but also the expected overall trend, in which higher $N_{res}$ exhibited a leftward shift phasor value, indicative of longer fluorescence lifetimes and increased $R_E$ (Supplementary Fig. 8c). To rule out intermolecular FRET artifacts, control experiments were performed using single-cysteine NHA9$_{A221C}$ mutants labelled either with the donor dye alone or with a mixture of donor and acceptor dyes. These samples were mixed with unlabelled protein at the same concentration as in the droplet assays. The fluorescence lifetime of the donor-only sample was indistinguishable from that of the donor-acceptor mixture, confirming the absence of intermolecular FRET (Fig. 4b). To quantify the $R_E$ values, we fitted the donor fluorescence lifetime decay curves using the Gaussian chain model according to Eq. 11 (see "Methods" for details). Our analysis yielded a scaling exponent of $v = 0.65 \pm 0.08$ for NHA9 in the droplet phase (Fig. 4c), compared to $v = 0.33 \pm 0.08$ in the nanocluster state (Fig. 2e), suggesting that the FG domain undergoes significant chain expansion within phase-separated droplets. These findings indicate a progressive expansion of the NHA9 FG domain as it transitions from a dilute monomeric state to an oligomeric and eventually a dense condensate state.

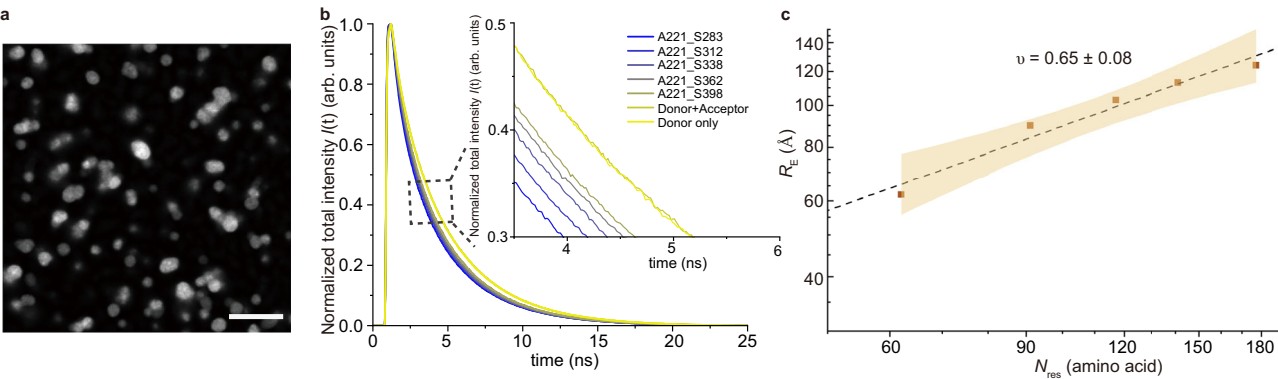

**Fig. 4 | Probing dimensions of NHA9 FG domain by FLIM-FRET for in vitro reconstituted condensates. a** Images of NHA9 droplets resting on coverslips. Scale bar: 10 µm. **b** Normalised donor fluorescence decays within droplets of five NHA9 variants labelled by Alexa594-LD655 dye pair. Fluorescence decays of single mutant NHA9$_{A221C}$ labelled with either Donor only or a Donor+Acceptor mixture were also included to confirm the absence of intermolecular FRET. **c** $R_E$ extracted from lifetime-based fitting and scaling law of the NHA9 FG domain within in vitro reconstituted condensates. $v$ is presented as the fitted value ± the standard error of the estimate. The yellow shaded area indicates the 95% confidence interval of the fit. Source data are provided as a Source Data file.

## Probing NHA9 FG domain conformation in cells

The cellular interior presents a highly crowded and heterogeneous environment, markedly distinct from the simplified conditions typically used in in vitro protein studies[52,53]. Therefore, directly probing the conformational dimensions of the NHA9 FG domain within cellular condensates remains both important and technically compelling. Given that fluorophore concentrations in cells are difficult to quantify and compare between cells—and that fluorescence lifetime is independent of fluorophore concentration, acceptor emission, and instrumental fluctuations—we again employed ensemble FLIM-FRET for this study. Previously, we developed a method combining FLIM-FRET with a site-specific synthetic biology approach to probe distance distributions of disordered nucleoporins within nuclear pore complexes in live cells[40]. In this approach, FLIM-FRET is further complemented by acceptor photobleaching, enabling precise measurement of donor-only fluorescence lifetimes and background signals.

Here, we applied this methodology to investigate the conformational dimensions of the FG domain of NHA9 with the GLEBS domain within transcriptional condensates in cells. Briefly, to achieve site-specific labelling, we employed genetic code expansion (GCE) to incorporate clickable non-canonical amino acids (ncAA) at defined positions (Fig. 5a). Specifically, codons in NHA9 variants were replaced with the amber stop codon (TAG), and the ncAA trans-cyclooct-2-en-l-lysine (TCO*A) was introduced to NHA9 site-specifically. Fluorophores bearing tetrazine groups were then added to permeabilised cells, and labelling was achieved via strain-promoted inverse-electron-demand Diels–Alder cycloaddition between TCO*A and the dye[54]. Since TCO*A incorporation can occur in any protein containing a TAG codon, resulting in nonspecific background labelling, we utilised our recently developed orthogonally translating organelles (OTOs) (Supplementary Fig. 9). These film-like compartments accumulate both orthogonal aminoacyl-tRNA synthetases (aaRS) and the mRNA of the protein of interest (POI), thereby achieving high recruitment and labelling specificity for the NHA9 mRNA during translation[55,56]. Furthermore, the use of small-molecule organic fluorophores for labelling ensured minimal linkage error and minimised disruption to the native structure and function of transcriptional condensates.

To confirm selective labelling of NHA9 by the OTO-GCE system, cells were co-transfected with plasmids encoding amber-mutated NHA9 and wild-type NHA9 fused to mEGFP. Upon treatment with TCO*A and LD655-H-tetrazine, colocalisation of LD655 and mEGFP-NHA9 signals was observed via fluorescence microscopy (Fig. 5b).

Additionally, immunofluorescence analysis confirmed NHA9 condensates exhibited stronger colocalisation with histone H3 lysine 27 acetylation (H3K27ac), but less colocalisation with histone H3 trimethyl K9 (H3K9me3), consistent with previous reports[29] (Fig. 5c, Supplementary Fig. 11). No labelling was observed when TCO*A was replaced with a non-reactive BOC ncAA. These results verify the presence of TCO*A and that the fluorophore was selectively incorporated into NHA9 condensates.

To accurately assess the conformational dimensions of the NHA9 FG domain, intermolecular FRET between different NHA9 molecules needed to be excluded. This was achieved by co-transfecting cells with a mixture of wild-type and TAG-mutated NHA9 plasmids to ensure a low concentration of labelled protein within condensates (Supplementary Fig. 10). Condensates were selected based on low acceptor intensity per pixel (excited at 660 nm) to minimise the likelihood of intermolecular FRET (Supplementary Fig. 12b). To further confirm the absence of intermolecular FRET, we expressed NHA9$_{A221TAG}$ in combination with wild-type NHA9, followed by treatment with a mixture of donor and acceptor dyes. As a result, each NHA9 molecule could be labelled with either a donor or an acceptor dye, but not both. Donor lifetimes were measured before and after acceptor photobleaching. An increase in donor lifetime post-photobleaching would indicate intermolecular FRET, whereas unchanged lifetimes would confirm its absence. We observed no change in donor fluorescence lifetime after acceptor photobleaching, indicating that intermolecular FRET did not occur (Fig. 5d, e).

We examined five dual-labelled FG-domain variants within cellular NHA9 condensates. Data were collected from around 30 cells per variant to account for cellular heterogeneity. Analysis of donor fluorescence lifetime decay curves revealed differences in FRET efficiency among the variants (Fig. 5f). These differences were further visualised using phasor plot analysis (Supplementary Fig. 12c), which confirmed the observed trend. To extract the apparent Flory scaling exponent ($v$), we performed global fitting of the decay curves using the Gaussian chain model according to Eq. 11 (see "Methods" for details), yielding $v = 0.61 ± 0.01$ (Fig. 5g). Additionally, a mutant-by-mutant analysis without assuming a specific model yielded $v = 0.63 ± 0.25$, in good agreement with the global fit (Supplementary Fig. 12d). We also measured three NHA9-derived mutants (A221-S283, A221-S312 and A221-S362) lacking the GLEBS domain. The measured distances were comparable to those observed in constructs containing the GLEBS domain (Supplementary Fig. 13), confirming that the scaling behaviour of the FG domain in cells is independent of the GLEBS domain.

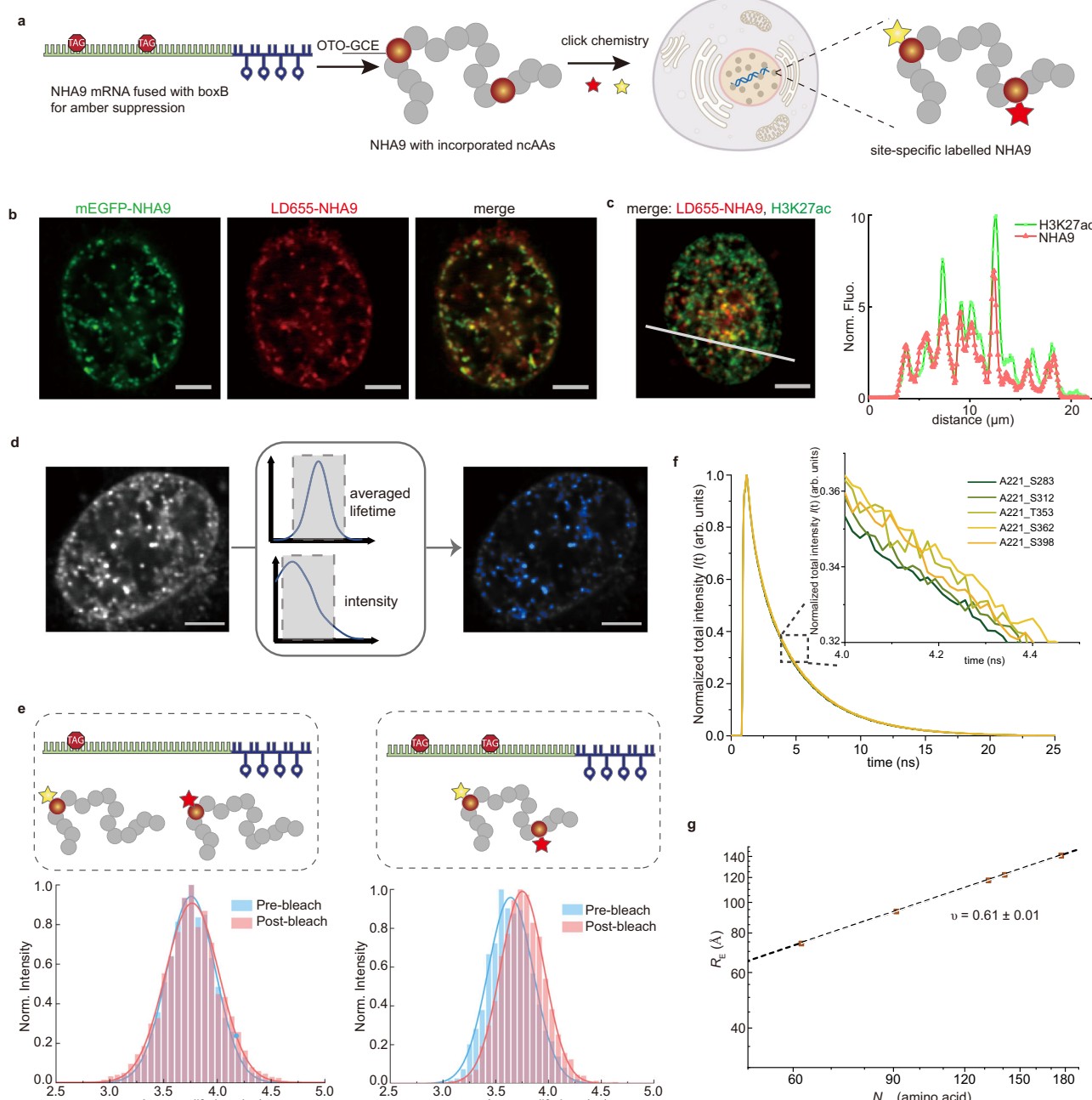

**Fig. 5 | Labelling and probing dimensions of NHA9 FG domain by FLIM-FRET in cells. a** Schematic of site-specific labelling of NHA9 in cells. NHA9 mRNA was fused with boxB loops, thus the mRNA was recruited into the orthogonally translating organelle genetic code expansion (OTO-GCE) system. ncAAs (non-canonical amino acids) were incorporated into NHA9 site-specifically, and a pair of FRET dyes (represented by a five-pointed star) was attached randomly to NHA9 by click chemistry after adding dyes to cells. **b** Imaging for mEGFP-NHA9 fusion (green) and NHA9 labelled by LD655 (red) via OTO-GCE to assay their colocalisation. **c** Left: Imaging for immunofluorescence of H3K27ac (green, using anti-H3K27ac antibody) and NHA9 labelled by LD655 (red) via OTO-GCE to assay their colocalisation. Right: Line scan along the white line in the magnified images shown in the left image. **d** Schematic of the FLIM-FRET analysis pipeline. NHA9 condensates were labelled by FRET dye pair via OTO-GCE, followed by acquiring donor decay profiles cell-by-cell. Nuclear condensates were selected as a region of interest to extract donor decay profiles afterwards. **e** Acceptor photobleaching assays for both single-mutant and double-mutant NHA9 labelled with FRET dye pair. The averaged donor lifetime distributions were acquired before and after acceptor photobleaching to infer FRET events exist or not. **f** Normalised donor fluorescence decays of five NHA9 variants labelled by FRET dye pair in cells. **g** $R_E$ extracted from lifetime-based fitting and scaling law of the NHA9 FG domain in cells from global fitting. $v$ is presented as the fitted value ± the standard error of the estimate. Scale bars: 5 µm. Cell image in Fig. 5a created with BioRender. Lemke, E (2025) [https://BioRender.com/e1pix2h]. Source data are provided as a Source Data file.

## Molecular simulations confirm NHA9's IDP expansion within the nanocluster and reveal non-stoichiometric micelle-like structures

To obtain a more detailed understanding of protein conformations, we performed molecular dynamics simulations of NHA9. To maintain consistency with the in vitro experiments, simulations were performed using NHA9 lacking the GLEBS domain. Residue-level coarse-grained simulations have proven effective in describing protein phase behaviour and capturing trends in experimental phase separation propensities. We employed the HPS-Urry force field, a model specifically

developed for implicit solvent simulations of IDPs[57]. The secondary and tertiary structures of HOXA9 were constrained using an elastic network model. Even though the HPS-Urry force field and the use of the elastic network do not accurately capture side-chain geometry and folded-domain interactions, they have nonetheless been shown to provide qualitatively reliable results in IDPs' self-assembly[58–61]. Simulations were conducted with varying numbers of NHA9 chains (ranging from 1 to 50) to investigate individual protein conformations across clusters of different sizes. We calculated the mean distances ($R_E$) between residues which were fluorescently labelled in the FRET experiments (Fig. 6a). These labelled residues, all located within the disordered FG domain of NHA9, exhibited trends consistent with experimental observations: Single NHA9 chains adopted highly collapsed conformations, whereas inter-residue distances increased with cluster size, indicating protein expansion in the nanocluster and condensate regimes. This behaviour is consistent with that observed for other IDPs[62].

By fitting the data presented in Fig. 6a, we computed the apparent Flory exponents for our simulated systems (Fig. 6b). For the single-chain system, we obtained a scaling exponent of $v = 0.20 \pm 0.01$, closely matching the experimentally measured value of $v = 0.24 \pm 0.02$. For cluster sizes of 5 and 10 chains, the exponents increased to $v = 0.32 \pm 0.01$ and $v = 0.37 \pm 0.01$, respectively. Beyond these sizes, $v$ continued to rise, reaching a value of $v = 0.46 \pm 0.004$ for the largest simulated cluster of 50 chains. These values are consistent with those obtained experimentally in the subsaturated regime, where nanoclusters form ($v = 0.33 \pm 0.08$). Due to computational limitations, we were unable to simulate sufficiently large systems to observe further expanded IDPs. Moreover, stretching of the shortest segment involved in the FRET measurements (62 amino acids) appeared to saturate at a cluster size of 30 chains, whereas longer segments continued to exhibit some degree of expansion as the cluster size increased from 30 to 50 chains. This suggests that further increases in cluster size would lead to further expansion of the IDP block, and increase the observed value of $v$. The heteropolymer nature of proteins may play a role here, as evidenced by heterogeneous structures recently reported in condensates as well as the approximate nature of coarse-grained force fields[63].

Our simulations captured the expansion behaviour of the IDP block of NHA9 in line with the available experimental data, therefore, we aimed to extend our analysis to the behaviour of the HOXA9 block. Snapshots from the simulated trajectories clearly show that the structured DBD preferentially localises near the surface of the nanoclusters (Fig. 6c, Supplementary Fig. 14). This localisation can be explained by the abundance of charged residues and the high net positive charge of the DBD (+8e, or +0.14e per residue), rendering it highly hydrophilic and leading to repulsive interactions with the hydrophobic IDP block, which also carries a slight positive charge (+0.02e per residue). To quantify this localisation bias, we computed density profiles as a function of distance from the cluster's centre of mass (COM) for both the IDP and HOXA9 blocks of NHA9 (Fig. 6d, Supplementary Figs. 15, 16). The results show that IDP residues are enriched near the cluster's centre, whereas the HOXA9 block exhibits a strong preference for localisation near the interface, resembling micelle-like structures. This behaviour aligns with the observed differences in hydrophobicity and net charge between the two domains and suggests a propensity for microstructure formation within the clusters. Additionally, the FG domain 2 was preferentially located near the centre of the condensate, while the FG domain 1, positioned at the chain end, was enriched at the interface (Supplementary Fig. 17 and Fig. 18), consistent with previous observations in polymer droplets[64]. This segregation becomes less pronounced as the cluster grows and both FG domains migrate to the centre of the condensate. Our findings also indicate that the DBD has a strong propensity to be at least partially in contact with the solvent, suggesting the preservation of its

native structure, which allows for DNA binding at the surface of the clusters. We also computed the intermolecular residue-residue contact maps, as shown in Supplementary Fig. 19. The analysis revealed an extensive network of contacts between the mostly hydrophobic residues in the IDP. This suggests that nanocluster formation is mediated mostly by hydrophobic IDP-IDP interactions, consistent with smFRET measurements under 1,6-hexanediol conditions (Supplementary Fig. 7). By contrast, the HOXA9 block forms, on average, fewer and more specific contacts.

We also performed simulations of NHA9, including the GLEBS domain to evaluate its role in nanocluster formation. Simulations with 10 chains were conducted, and density profiles of the IDP and HOXA9 blocks were derived from the trajectories (Supplementary Fig. 20). The density profiles of NHA9 with and without the GLEBS domain were nearly identical, showing only a slight increase in cluster size for the construct containing the GLEBS domain, attributable to the additional 57 residues. Furthermore, both $R_E$ values and $v$ values were highly similar (Supplementary Fig. 21). Together, these results indicate that the GLEBS domain does not significantly influence NHA9 assembly.

## Discussion

Transcriptional condensates play a critical role in regulating gene expression by organising TFs and the associated transcription machinery into functional compartments[7,11,65]. Increasing evidence indicates that many proteins prone to condensation are associated with the early formation of large nanoclusters, both below and above their macroscopic phase separation thresholds[32,34]. In this study, we show that NHA9 forms nanoclusters at subsaturated concentrations, below the critical concentration for phase separation. Furthermore, we demonstrate that DNA binding modulates NHA9 nanocluster formation, suggesting a regulatory mechanism in which chromatin interactions can influence protein assembly dynamics. Given that NHA9 nanoclusters form at concentrations below 100 nM, it is likely that they arise under physiological conditions and contribute to transcriptional regulation through a mechanism distinct from classical macroscopic phase-separated condensates, similar to the reported gene-regulatory role of the TF Mig1 clusters[35].

The distinct assembly states of NHA9 are primarily driven by its long, intrinsically disordered FG domain. However, the precise molecular mechanisms underlying NHA9 self-assembly and its structural dynamics remain poorly understood. To elucidate these mechanisms, we systematically investigated NHA9 behaviour across a range of protein concentrations. Our data revealed a conformational transition in the FG domain, progressing from a dilute monomeric state to an oligomeric state, and ultimately to a dense phase (Fig. 7a, b). Both inter- and intramolecular interactions mediate these transitions. In the dense phase, extensive protein-protein interactions promote chain expansion, whereas in the dilute phase, intramolecular contacts dominate, resulting in a more compact conformation that minimises free energy. Surprisingly, our results suggest that the oligomeric state is structurally similar to micelles, with the hydrophilic DBD domain located near the surface—a feature also predicted by computational simulations for Nanog, a master TF[66]. Experimentally, the number of chains in the structure increases with overall concentration, hinting at morphologies that are not typical core-shell micelles. The NHA9 micelles have different appearances from those micelles recently suggested for TDP43 and SRSF proteins, which present fixed-size morphologies on electron micrographs[67]. NHA9 exhibits a weaker amphiphilic block copolymer character compared to TDP-43 and SRSF, potentially explaining the observed differences in micellar appearance. Micellar systems usually exhibit a critical micelle concentration (CMC), above which most of the chains are part of micelles. Lack of a clear CMC in our system could be explained by a plethora of underlying effects, for instance, the existence of multiple competing micellar structures, i.e., spherical or worm-like chains. Our simulation

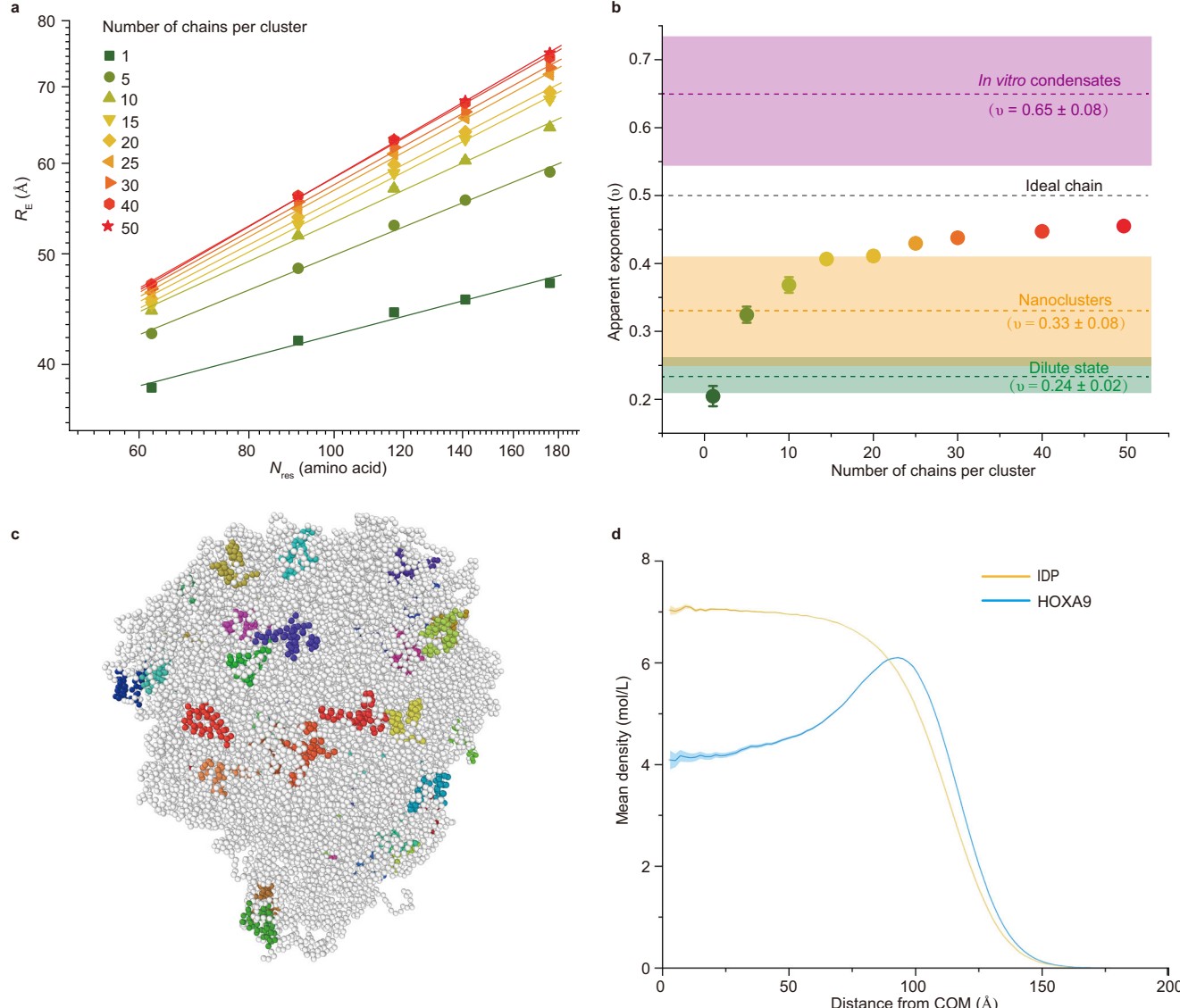

**Fig. 6 | Coarse-grained simulations of NHA9 in dilute states and nanoclusters.**
**a** Mean end-to-end distances ($R_E$) vs sequence distance ($N_{res}$) for clusters of different sizes extracted from coarse-grained simulations. The solid lines show scaling law fitting. Data are presented as mean values ± standard error of the mean (SEM). The number of frames varies for each data point and is provided in Supplementary Table 4. **b** Apparent Flory exponent ($\nu$) as a function of the number of chains in cluster, obtained by fitting the data points shown in Fig. 6a. Circular dots represent clusters with different numbers of chains, and the corresponding $\nu$ values are shown as mean ± the square root of the diagonal term of the covariance matrix of the fit corresponding to $\nu$. Dashed lines represent scaling exponent values derived from experimental measurements across different assembly states. Coloured

rectangles represent experimental results, corresponding to fitted scaling values ± standard error of the estimate. Green, orange, and purple indicate the dilute, nanocluster, and in vitro condensate states, respectively. **c** Snapshot of a cluster containing 50 NHA9 chains. The coloured amino acids represent the helices in the DNA-binding domain HOXA9 and the remaining amino acids are shown in white. Snapshot was made with OVITO[90]. **d** Mean density of the intrinsically disordered protein (IDP) block and the HOXA9 block in NHA9, plotted as a function of the distance from the cluster centre of mass (COM). Units are moles of amino acids per liter. Results correspond to the cluster containing 50 chains. Data are presented as mean values ± SEM, with the SEM shown as a shaded region (n = 8028 frames). Source data are provided as a Source Data file.

results nevertheless indicate that NHA9 nanoclusters possess a distinct substructure wherein the DBD is oriented toward the exterior despite their inherent heterogeneity and stochastic assembly. This structural feature appears intrinsically encoded within the oncofusion block copolymer NHA9 sequence properties. Unlike typical surfactants, we observe phase separation at sufficiently high concentrations. In block copolymers, a competition between phase separation and micellisation under weakly amphiphilic conditions can drive transitions between micellar and phase-separated regimes[68,69]. Consequently, the presence of phase separation at higher concentrations does not preclude the formation of biologically functional micelles at low

concentrations. While the literature sometimes distinguishes between the self-assembly of surfactants and microphases, they are driven by the same underlying physics. Dense protein phases could similarly show microphases, structure, or heterogeneities, which may be functionally relevant[63]. These results, along with other recent reports of micellar systems[67,70], highlight the need to further explore the functional significance of nano assemblies and microstructure.

In the cellular environment, TF condensates function as dynamic hubs that recruit components of the transcriptional machinery—including mediator, general TFs, and RNA polymerase II—to facilitate

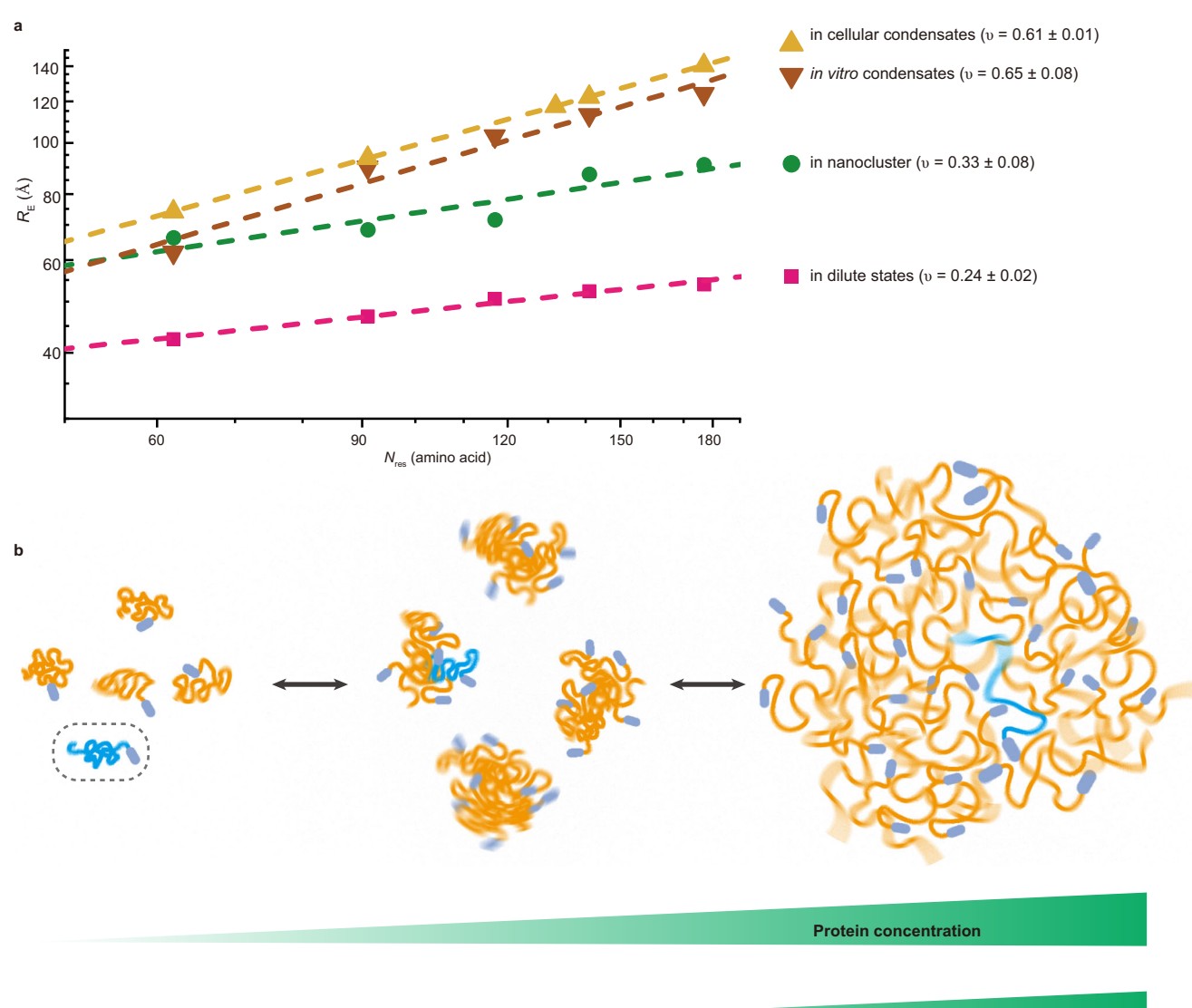

**Fig. 7 | Concentration-dependent self-assembly of NHA9. a** Summary of scaling exponents of FG domain of NHA9 under different assembly states. *v* is presented as fitted values ± the standard error of the estimate. The disordered FG domain is the driving force behind NHA9 self-assembly, and our data revealed that the FG domain became increasingly expanded from the dilute state to nanoclusters and condensates. **b** Conceptual scheme illustrating NHA9 FG domain expansion within different assembly states. Source data are provided as a Source Data file.

transcription initiation. Our in situ measurements reveal that the FG domain of NHA9 adopts a more extended conformation in cellular phase-separated condensates ($v = 0.61 \pm 0.01$), which is consistent with in vitro condensates ($v = 0.65 \pm 0.08$). This conformation results from self-association and may also enhance interactions between NHA9 and components of the transcription machinery. While the cellular FLIM-FRET measurements provided valuable insights, they are not without limitations that may influence the results. First, the data were obtained from transiently transfected cells, which results in NHA9 over-expression and does not reflect endogenous expression levels. Consequently, the observed NHA9 condensates often exceed 1 µm in diameter, potentially diverging from physiological conditions. TFs expressed at endogenous levels generally form numerous smaller transcriptional hubs or condensates; for example, HSF1 forms condensates with diameters of approximately 300 nm[10,18]. Recent studies have further shown that such localised, high-concentration transcriptional hubs—typically containing fewer than 100 molecules and exhibiting dwell times of approximately 1–2 minutes—are critical for effective transcriptional activation[71]. Furthermore, smaller condensates with signals near background levels were excluded in our FLIM-FRET analyses, limiting our observations to larger structures. Second, the GCE approach used for site-specific labelling introduces an in-frame stop codon, potentially resulting in truncated NHA9 variants that may influence condensate behaviour. Third, although mild plasma membrane permeabilisation facilitated the use of hydrophilic dyes with reduced nonspecific binding, some background signal from free dyes likely remained, potentially reducing the signal-to-noise ratio. Future studies should aim to optimize GCE strategies to prevent protein truncation and enable endogenous-level labelling, as well as to employ fluorogenic, cell-permeable dyes to improve imaging specificity and quality, thereby enabling the investigation of conformational ensembles within nanoclusters. Despite these technical constraints, our in situ scaling measurements avoided using large fluorescent labels, such as GFP or self-labelling proteins, thereby minimising potential perturbation of NHA9 condensates in cells. Most importantly, the data support the conclusion that the disordered FG domain of NHA9 adopts a similarly extended conformation in both in vitro and cellular environments.

Our study provides a molecular-level understanding of biomolecular conformational shifts during the condensation process. At low concentrations, proteins exist as individual molecules in the dilute phase. As concentration increases, they begin assembling into nanoclusters, and above the saturation threshold, these clusters coalesce into a condensed phase. Importantly, our findings reveal that this assembly process is accompanied by a continuous conformational transition from a compact chain in the monomeric state to expanded ones in the oligomeric state, culminating in more stretched chains in a densely packed phase-separated condensate. For IDPs with varying properties of blocky segments, scaling law measurements and chain expansion should be assessed separately for each segment. Furthermore, we found that the oncofusion protein NHA9 exhibits characteristics analogous to a weakly amphiphilic diblock polymer. Specifically, the disordered FG-rich domain in NHA9 resembles a hydrophobic block, while the DBD resembles a hydrophilic block. Notably, this weak amphiphilic architecture enables NHA9 to self-assemble into non-core-shell micelles, with the DBD preferentially oriented towards the exterior. Such a configuration is likely functionally significant, as it facilitates direct access of DNA to the DBD, thereby promoting DNA interactions, which could also orchestrate long-range chromatin interactions such as enhancer-promoter looping at oncogene loci. Since oncofusion TFs are typically formed from two blocks, an IDP block and a DBD block, their block copolymer–like properties may represent a common feature among oncofusion TFs. The formation of non-core-shell spherical micelles could serve as a general assembly principle underlying their oncogenic activity. As many eukaryotic TFs already obey block copolymer characteristics, we could expect the formation of non-core-shell spherical micelles also for those with weak amphiphilicity and a long IDP region. These insights contribute to a deeper understanding of the molecular principles governing transcriptional condensates and highlight the importance of structural transitions in regulating transcriptional activity.

## Methods

### Protein expression and purification

The human NHA9 protein lacking the Gle2-binding sequence (GLEBS; amino acids 157–203) was cloned into the bacterial expression vector pQE-14His-TEV by restriction cloning. Recombinant NHA9 was expressed in *Escherichia coli* strain BL21-AI. Cultures were grown in Luria-Bertani (LB) medium supplemented with 50 µg/mL kanamycin at 37 °C until reaching an optical density at 600 nm ($OD_{600}$) of approximately 0.6. Protein expression was induced with 1 mM isopropyl-β-d-thiogalactoside (IPTG) and 0.02% (w/v) arabinose at 18 °C. Following overnight expression, cells were harvested by centrifugation and resuspended in lysis buffer containing 6 M guanidine hydrochloride, 50 mM Tris-HCl, 1 M NaCl, 20 mM imidazole, and 2 mM 2-mercaptoethanol at pH 9.0. Cells were lysed by sonication, and the lysate was clarified by centrifugation at $19,000 \times g$ for 1 h at 4 °C. The supernatant was incubated with Ni-NTA agarose beads for 2 h at 4 °C. Beads were subsequently washed with buffer containing 4 M guanidine hydrochloride, 50 mM Tris-HCl, 2 M NaCl, 40 mM imidazole, and 2 mM 2-mercaptoethanol at pH 9.0. The his-tagged protein was eluted with buffer containing 2 M guanidine hydrochloride and 50 mM Tris-HCl at pH 8.0. Eluted proteins were dialysed against buffer containing 0.5 M guanidine hydrochloride and 50 mM Tris-HCl at pH 8.0 using dialysis membranes with a molecular weight cutoff (MWCO) of 6–8 kDa to remove imidazole. His-tags were cleaved overnight at room temperature with TEV protease. Typically, proteins precipitated following cleavage and were resolubilised in 4 M guanidine hydrochloride at pH 9.0. The solution was incubated again with Ni-NTA beads to remove uncleaved protein and the cleaved his-tag. The flow-through was concentrated and further purified by size-exclusion chromatography using a Superdex 75 column equilibrated with 4 M guanidine hydrochloride, 50 mM Tris-HCl, 2 M NaCl, and 0.2 mM TCEP at pH 9.0. Fractions were analysed by SDS-PAGE using 4–12% gradient gels with MOPS running buffer and stained with Coomassie Brilliant Blue. Pure fractions were pooled and concentrated to approximately 15 mg/mL using 3-kDa MWCO centrifugal filters (Merck Millipore), and protein concentrations were determined using the BCA Protein Assay Kit (Thermo Fisher). Final protein preparations were flash-frozen in liquid nitrogen and stored at −80 °C.

For constructs containing a TGT codon (encoding cysteine) at the second residue and a TAG amber codon at position A221 (to incorporate p-acetylphenylalanine, AcF, via amber suppression), sequences were cloned into the pQE-14His-TEV vector. All constructs were co-transformed with the pEvol plasmid, encoding an evolved aminoacyl-tRNA synthetase specific for AcF and its cognate tRNA, into *Escherichia coli* BL21-AI cells[72]. Cultures were grown in LB medium supplemented with 50 µg/mL kanamycin and 33 µg/mL chloramphenicol at 37 °C until $OD_{600}$ reached 0.2, at which point 1 mM AcF was added. Induction and purification followed the same procedure as described for the non-TAG constructs.

### NHA9 FG domain labelling in vitro

**NHA9 labelling for FLIM-FRET measurements in reconstituted condensates.** Purified NHA9 containing double cysteine mutations was reduced with 10 mM 1,4-dithiothreitol (DTT) at room temperature (RT) prior to labelling with thiol-reactive dyes. Following reduction, the protein was buffer-exchanged into maleimide labelling buffer (4 M guanidine hydrochloride, 1× PBS, 0.1 mM EDTA, 0.2 mM TCEP, pH 7.0) using a centrifugal filter with a 3 kDa molecular weight cutoff (MWCO; Amicon). Labelling was performed overnight at 4 °C using Alexa Fluor 594 maleimide (A10256, Thermo Fisher) and LD655-maleimide (Lumidyne Technologies) at a 1:2 dye-to-protein molar ratio. The reaction was quenched by adding DTT to a final concentration of 10 mM. Free dyes were removed via buffer exchange followed by gel filtration using a Superdex 75 column.

**NHA9 labelling for smFRET measurements.** For single-molecule FRET (smFRET) labelling, NHA9 variants containing a single cysteine and a TAG amber codon were site-specifically labelled with Alexa Fluor 594-maleimide at the cysteine residue (acceptor) and Alexa Fluor 488-hydroxylamine at the AcF residue (donor), following established protocols[72]. Briefly, samples were exchanged into oxime labelling buffer (50 mM sodium acetate-HCl, pH 4.0, 150 mM NaCl, 4 M guanidine hydrochloride, 100 mM aniline). A total of 50 µL of 200 µM protein was reacted with 50 µL of 1 mM Alexa Fluor 488-hydroxylamine (a fivefold molar excess of dye) for 24 h at 37 °C. After the reaction, samples were exchanged into maleimide labelling buffer through three rounds of buffer exchange using a 3 kDa MWCO centrifugal filter (Amicon), followed by reduction with 10 mM DTT. DTT was then removed by five additional rounds of buffer exchange. The freshly prepared protein was subsequently labelled with Alexa Fluor 594 maleimide (1:2 dye-to-protein molar ratio) overnight at 4 °C. The reaction was quenched with 10 mM DTT, and unreacted dye was removed by buffer exchange and gel filtration using a Superdex 75 column.

### Circular dichroism spectroscopy

CD spectra were recorded using a JASCO J-1500 spectrometer (JASCO Corporation, Tokyo, Japan) over the wavelength range of 190–260 nm. The instrument was operated with the following parameters: scan rate, 100 nm/min; cell path length, 1 mm; step resolution, 1 nm; bandwidth, 1 nm; data integration time, 1 s; and six accumulations at RT. To enhance buffer transparency, CD measurements were performed in 10 mM Tris-HCl (pH 7.4) supplemented with 80 mM guanidine hydrochloride at a protein concentration of 0.2 mg/mL.

## Protein Thermal Stability Studies

Thermal unfolding of NHA9 was analyzed using a Prometheus NT.48 instrument (NanoTemper Technologies, Germany). Samples containing 1.2 µM NHA9 were measured either in a buffer containing 50 mM Tris-HCl (pH 7.4), 2 M guanidine hydrochloride or in a buffer containing 10 mM Tris-HCl (pH 7.4), 75 mM NaCl, 80 mM guanidine hydrochloride. Thermal denaturation experiments were conducted with a linear thermal ramp from 20 to 95 °C at a heating rate of 1 °C/min with 100% excitation power. The fluorescence intensity (F330) was plotted as a function of temperature. All experiments were performed in triplicate.

## Electrophoretic mobility shift assay (EMSA)

Varying concentrations of NHA9 (up to 2 µM) were incubated with 10 nM Cy5-labelled double-stranded DNA in a buffer containing 10 mM Tris-HCl (pH 7.4), 75 mM NaCl, 1 mM EDTA, and 3% glycerol. For the DNA fragment containing the HOXA9 binding site, we used the sequence 5′-ACTCTATGATTTACGACGCT-3′ (HOXA9 binding site, TTTAC) as previously reported[30]. The corresponding shuffled sequence is 5′-CTATATGCTATCGTGTAACC-3′. After mixing, the protein-DNA complexes were incubated for 5 min at room temperature. Samples were then resolved on a 6% native polyacrylamide gel prepared in 1× TBE buffer and electrophoresed at a constant voltage of 100 V for 40 min. Fluorescently labelled DNA-protein complexes were visualised using a ChemiDoc MP imaging system (Bio-Rad). The oligonucleotides were ordered from Genewiz (NJ, US). Band intensities were measured using Fiji[73].

## Turbidity measurement

Purified NHA9 in 2 M guanidine hydrochloride was rapidly mixed with 24 µL of buffer (50 mM Tris-HCl, pH 7.4, 150 mM NaCl). The resulting mixture (total volume: 25 µL) was immediately transferred to a clear-bottom 384-well plate, and turbidity was assessed by measuring absorbance at 340 nm using a SpectraMax iD5 plate reader. All measurements were performed in triplicate for each construct under the specified conditions.

## Cell culture, transient transfection and labelling

The HeLa Kyoto cell line (RRID: CVCL_1922), kindly provided by Martin Beck's laboratory, was cultured in DMEM supplemented with 1 g/L glucose (Gibco, 31885023), 9% foetal bovine serum (FBS; Sigma, F7524), 1% penicillin-streptomycin (Thermo Fisher, 15140-122), and 1% L-glutamine (Thermo Fisher, 25030-081) at 37 °C in a humidified incubator with 5% CO2. Cells were routinely tested and confirmed to be free of mycoplasma contamination. Constructs used for transient transfection were cloned using either restriction cloning or Gibson assembly.

To introduce NHA9 transcriptional condensates, HeLa cells were transiently transfected with plasmids encoding NHA9. Co-transfection with plasmids encoding TAG-mutated NHA9 and the OTO-GCE system enabled site-specific labelling. To minimise intermolecular FRET signals caused by heterogeneity of transient expression, wild-type NHA9 was co-expressed in excess. For this purpose, a bicistronic construct (pBI_Flag::NHA9TAG::HA(boxB)_Flag::NHA9::HA::P2A-T2A::NHA9) was designed using the pBI_CMV vector and self-cleaving peptides to promote higher expression of wild-type NHA9 relative to the TAG-mutant. Additional plasmids encoding wild-type NHA9 were also co-transfected to further increase the expression ratio. For FLIM-FRET experiments, 150,000 HeLa cells were seeded per 35 mm µ-dish (ibidi, 81158) one day prior to transfection. Transfection was performed using jetPRIME reagent (Polyplus-transfection) with 1.8 µg total DNA, 150 µL jetPRIME buffer, and 4 µL jetPRIME reagent per dish. A 1:1:1 mass ratio of the following plasmids was used:

- pBI_Flag::NHA9::HA
- pBI_Flag::NHA9$^{TAG}$::HA(boxB)_Flag::NHA9::HA::P2AT2A::NHA9

- pcDNA3.1_TOM20$_{1-70}$::FUS$_{1-478}$::V5::Myc::4xλ$_{N22}$::NES::PylRS$^{Y306A,Y384F}$_U6-tRNA$^{Pyl,CUA}$

A list of all plasmids used is provided in Supplementary Table 1. The organelle plasmid (pcDNA3.1_TOM20$_{1-70}$::FUS$_{1-478}$::V5::Myc::4xλ$_{N22}$::NES::PylRS$^{Y306A,Y384F}$_U6-tRNA$^{Pyl,CUA}$) was adapted from previous studies[40,56]. Four to five hours post-transfection, the culture medium was replaced with fresh medium containing 60 µM trans-cyclooct-2-en-l-lysine (TCO*A; SciChem) for in-cell labelling, or tert-butyloxycarbonyl-l-lysine (BOC; Iris Biotech) for control experiments. Media were also supplemented with 10 mM HEPES (pH 7.25). Stock solutions of 100 mM for all ncAAs were prepared in 15% (v/v) DMSO/0.2 M NaOH, following established protocols[54].

Twenty to twenty-four hours after transfection, cells were washed twice with transport buffer (TB: 20 mM HEPES, 110 mM KOAc, 5 mM NaOAc, 2 mM MgOAc, 1 mM EGTA, 2 mM DTT, pH 7.3 adjusted with KOH), supplemented with 5 mg/mL PEG6000 to prevent osmotic shock. To minimise interference from cellular autofluorescence in the green spectral range, we employed the AF594-H-tetrazine and LD655-H-tetrazine dye pair. Cellular delivery of the dyes was achieved via mild digitonin treatment, which permeabilises the plasma membrane while preserving nuclear envelope integrity. Specifically, cells were then permeabilised by incubation with 40 µg/mL digitonin (AppliChem, A1905) for 10 min at room temperature. Following two washes with TB, cells were incubated in a dye solution containing 33.3 nM Alexa Fluor 594-H-tetrazine (Click Chemistry Tools) and 66.6 nM LD655-H-tetrazine (Lumidyne Technologies) in TB for 5 min at room temperature. A 1:2 donor-to-acceptor dye ratio was used to reduce the donor-only population[51]. After labelling, cells were washed in TB and incubated at 37 °C for 30 min to remove residual dyes. Cells were used for imaging within a 3-hour window post-labelling.

## Western blot assay

HeLa cells were seeded in a 6-well plate 15–20 h prior to transfection to reach 70–80% confluency at the time of transfection. Each well was transfected with 3 µg of the plasmid [pBI_Flag::NHA9TAG::HA(boxB)_Flag::NHA9::HA::P2A-T2A::NHA9] using jetPRIME (Polyplus-transfection). Twenty to twenty-four hours after medium replacement, cells were harvested and lysed in 40 µL RIPA buffer (150 mM NaCl, 1.0% Triton X-100, 0.5% sodium deoxycholate, 0.1% SDS, 50 mM Tris-HCl, pH 8.0) supplemented with Complete Protease Inhibitor Cocktail (Roche, 11873580001), 1 mM MgCl2, and Sm nuclease (Protein Production Core Facility, IMB Mainz). Protein lysates were resolved by SDS-PAGE using NuPAGE 4–12% Bis-Tris gels with NuPAGE MOPS SDS Running Buffer (Thermo Fisher Scientific), and transferred to nitrocellulose membranes (0.2 µm pore size; Trans-Blot Turbo Midi, Bio-Rad, 1704159) using the Trans-Blot Turbo Transfer System (Bio-Rad). Membranes were blocked for 1 hour at room temperature (RT) in 5% low-fat milk (Carl Roth, T145.3) prepared in PBS. Blots were incubated overnight at 4 °C with a mouse anti-HA primary antibody (Sigma, H9658, 1:1000 dilution) in blocking solution. The following day, membranes were washed with PBST (PBS containing 0.2% Tween-20), then incubated with a secondary antibody (goat anti-mouse HRP-conjugated, 1:10,000 dilution; Jackson ImmunoResearch, 715-035-150) for 1 hour at RT. After final washes with PBST, membranes were incubated for 1 min with enhanced chemiluminescence substrate (ECL; Cytiva, RPN2106). Signal detection was performed using a ChemiDoc MP imaging system (Bio-Rad).

## Fixed cell immunofluorescence

HeLa cells were transfected with the pcDNA3.1_mEGFP::4×GGS::NHA9 plasmid (where 4×GGS serves as a flexible linker), plasmids encoding TAG-mutated NHA9 and the OTO-GCE system for colocalisation assay

between amber-mutated NHA9 and wild-type NHA9 fused to mEGFP (Fig. 5b). For histone immunofluorescence imaging in Fig. 5c, HeLa cells were transfected with plasmids encoding TAG-mutated NHA9 and the OTO-GCE system and incubated for 24 h prior to fixation. Cells were firstly labelled with 100 nM LD655-H-tetrazine (Lumidyne Technologies) as described before and then fixed with 2% paraformaldehyde in PBS for 10 min at room temperature (RT), permeabilised with 0.5% Triton X-100 in PBS for 15 min at RT, and rinsed twice with PBS before blocking. Blocking was performed using 3% bovine serum albumin (BSA) in PBS for 90 min at RT. Cells were then incubated overnight at 4 °C with a primary antibody against H3K27Ac (histone H3 lysine 27 acetylation; 1:1000 dilution, Abcam, ab4729) and H3K9me3 (histone H3 tri methyl K9; 1:1000 dilution, Abcam, ab8898) diluted in blocking buffer. The following day, samples were washed with PBS and incubated with a secondary antibody (anti-rabbit, Alexa Fluor 488, 1:2000 dilution, ThermoFisher, A-11008) for 60 min at RT. After final rinses with PBS, fresh PBS was added to the wells, and cells were prepared for imaging.

### Single-molecule FRET measurements

Single-molecule fluorescence experiments were conducted using a custom-built multiparameter spectrometer centred around a high-numerical-aperture water-immersion objective (Nikon 60×, 1.27 NA) mounted on a z-translator[40]. Linearly polarised outputs from three picosecond pulsed laser diodes−485 nm (LDH-D-C-485, PicoQuant), 560 nm (LDH-D-TA-560, PicoQuant), and 660 nm (LDH-D-C-660, PicoQuant)−were passed through corresponding excitation filters (488/10, 560/14, and 661/11). Fluorescence emission was spatially filtered using a 100 μm pinhole, then split into parallel and perpendicular polarisation components. Each polarisation component was further spectrally separated into green, orange, and red channels using emission filters (525/50, 609/57, and 700/75, respectively) and focused onto single-photon counting detectors (green: MPD, PicoQuant; orange: PMA Hybrid 40, PicoQuant; red: τ-SPAD, PicoQuant). Laser pulses were alternated to probe the presence of acceptor fluorophores, with synchronisation controlled by a multichannel laser driver (Sepia II, PDL 828, PicoQuant)[74,75]. The entire setup was operated using SymPhoTime64 software (PicoQuant GmbH). Samples labelled with Alexa Fluor 488 and Alexa Fluor 594 were excited using the 485 nm laser at 55 μW and the 560 nm laser at 65 μW, both operating at a repetition rate of each 25 MHz. Measurements were performed at a sample concentration of ~50 pM in buffer containing 50 mM Tris-HCl (pH 7.4), 150 mM NaCl, 0.01% Tween-20, and freshly added 10 mM DTT. Photon detection was performed using a multichannel time-correlated single-photon counting module (HydraHarp 400, PicoQuant) with a time resolution of 16 ps. Polarisation correction factors were determined to be $L_1 = 0.16$ and $L_2 = 0$ (Eq. 9).

### Analysis of transfer efficiency histograms

Single-molecule fluorescence data were analysed using the PAM software[76]. Single-molecule events were identified using the All-Photon Burst Search algorithm with a threshold of 5 photons per 1000 μs time window and a minimum total photon count of 50 per burst. The FRET transfer efficiency ($E$) and stoichiometry ($S$) for each burst were calculated as follows:

$$E = \frac{I_A^D}{\gamma I_D^D + I_A^D} \tag{1}$$

$$S = \frac{\gamma I_D^D + I_A^D}{\gamma I_D^D + I_A^D + I_A^A} \tag{2}$$

where $I_A^D$ is the corrected intensity from the acceptor after donor excitation, $I_D^D$ is the corrected intensity from the donor after donor excitation and $I_A^A$ is the corrected intensity from the acceptor after acceptor excitation. Raw intensities were corrected for background, donor signal leakage into the acceptor channel ($\alpha = 0.086$), and direct excitation of the acceptor by the donor-exciting laser ($\delta = 0.046$). The $\gamma$ factor accounts for differences in quantum yield and detection efficiency between the donor and acceptor fluorophores ($\gamma = 0.323$).

To estimate the mean transfer efficiency, the transfer efficiency histograms were fitted with a 2D Gaussian function. For smFRET measurements of nanoclusters, stoichiometry values of nanocluster species were below 0.5 because of acceptor quenching. The Donor-only species and FRET species, indicated by blue rectangles, were individually selected for separate fitting using a 2D Gaussian function, as shown in Supplementary Fig. 4e. The fitted mean transfer efficiencies $\langle E \rangle$ values were applied to acquire the root-mean-square end-to-end distances $R_E = \sqrt{<r^2>}$ by numerically solving the following equation:

$$<E> = \int_0^\infty dr E(r) P(r) \tag{3}$$

$$E(r) = \frac{R_0^6}{R_0^6 + r^6} \tag{4}$$

$$P(r) = 4\pi r^2 \left(\frac{3}{2\pi\langle r^2\rangle}\right)^{\frac{3}{2}} e^{-\frac{3r^2}{2\langle r^2\rangle}} \tag{5}$$

where $R_0$ is the Förster radius with $R_0 = 5.6$ nm for a Alexa488/Alexa594 dye pair and $R_0 = 7.7$ nm for a Alexa594/LD655 dye pair according to previous reports[40,42]. For the distance distribution function between the donor and acceptor dyes, $P(r)$, we used a Gaussian chain model, as given by Eq. 5. In smFRET measurements the donor and acceptor anisotropies are low ($r < 0.3$), indicating the free rotation of dyes. Finally, the extracted $R_E$ values were used to determine the polymer scaling law factor by fitting to:

$$R_E = \rho_0 N_{res}^v \tag{6}$$

### Fluorescence correlation spectroscopy (FCS) measurements

FCS measurements were performed using the same setup as for smFRET experiments, employing a 660 nm laser (LDH-D-C-660, PicoQuant) operated in continuous-wave mode at a power of 50 μW with a 50 μm pinhole. Experiments were conducted with 10 nM LD655-labelled NHA9 protein mixed with varying concentrations of unlabelled NHA9 in buffer containing 50 mM Tris-HCl (pH 7.4), 150 mM NaCl, 0.01% Tween-20, and freshly added 10 mM DTT. A total of 120 measurements, each with a 10-second acquisition time, were collected for each condition. Data analysis was carried out using SymPhoTime 64 software. Autocorrelation curves were generated for lag times ranging from 0.0001 ms to 1000 ms and fitted using a standard diffusion model (Eq. 7), where $\omega_0/\omega_Z$ is the structure parameter, $\tau_D$ and $N$ represent the readout parameters of the fit.

$$G(\tau) = \frac{1}{N}\frac{1}{1+(\tau/\tau_D)}\sqrt{\frac{1}{1+(\tau/\tau_D)(\omega_0/\omega_Z)^2}} \tag{7}$$

$$D = \frac{k_B T}{6\pi\eta r} \tag{8}$$

The Stokes-Einstein equation (Eq. 8) describes how small spherical particles diffuse through a fluid with constant temperature by undergoing Brownian motion and can be used to calculate the particle's hydrodynamic radius. In Eq. 8, D is the translational diffusion coefficient, $k_B$ is Boltzmann's constant, $T$ is the absolute temperature, $\eta$ is the solvent viscosity, and $r$ is the hydrodynamic radius of the particle.

### In vitro droplet assay and FLIM−FRET measurements

Purified labelled and unlabelled NHA9 proteins were mixed at a 1:10,000 ratio in 2 M guanidine hydrochloride. A 1 µL aliquot of the mixed protein solution was rapidly diluted into 24 µL of assay buffer (50 mM Tris-HCl, 150 mM NaCl, 10 mM DTT) on a 15-well µ-slide (ibidi, 81507, Germany). The final concentrations were 10 µM unlabelled and 1 nM labelled NHA9. FLIM acquisition was initiated within the first 5 min after forming the droplets, as the condensates were more liquid during this time. Over time, the droplets hardened out, likely due to molecular ageing as observed also for Nup98 FG domains[77]. Samples were excited using a pulsed laser operating at 40 MHz with a power of 70 µW. Imaging was performed using SymPhoTime 64 software with the following parameters: pixel size of 200 nm, dwell time of 100 µs, image resolution of 256 × 256 pixels, and a time resolution of 16 ps. Fluorescence anisotropy values are presented in Supplementary Fig. 8a, b.

### Fluorescence recovery after photo bleaching (FRAP)

NHA9 droplets were prepared as described above. FRAP experiments were performed on a Visitron spinning-disk microscope (Visitron Systems GmbH) together with an ORCA-flash 4.0 camera C13440 CMOS camera (Hamamatsu Photonics). Bleaching was done with a 488 nm laser at 70% power. Time-lapse images were acquired over a 98 s time course after bleaching with a 2 s interval. Fluorescence intensities of regions of interest were corrected by unbleached control regions and then normalized to pre-bleached intensities of the regions of interest.

### FLIM-FRET for cell measurements

The same optical setup used for smFRET was employed for FLIM-FRET measurements, both in vitro (droplets) and in live cells. For cellular imaging, the following acquisition parameters were used: pixel size of 100 nm, image resolution of 256 × 256 pixels, pixel dwell time of 150 µs, and time resolution of 16 ps. To ensure proper sample quality, labelled cells were first imaged under 660 nm laser excitation to confirm the absence of cytoplasmic aggregates ("blobs"), which indicate the presence of insoluble NHA9 truncation products. The acceptor fluorescence intensity per pixel within nuclear condensates was also assessed under 660 nm excitation to evaluate expression levels and avoid overexpression, which could result in intermolecular FRET. Expression levels were further validated through acceptor photobleaching. A constant donor fluorescence lifetime before and after photobleaching confirmed the absence of intermolecular FRET (see Fig. 5e). Cells that met these criteria were subjected to FLIM imaging. First, imaging was performed using 560 nm laser excitation (30 µW, 40 MHz) for 5 min. Next, a 30-s acquisition was conducted using the 660 nm laser (30 µW, 40 MHz), followed by Acceptor photobleaching using the same laser at a higher power (300 µW, 40 MHz) for 2 min. Post bleaching, the cell was again imaged for 5 min under 560 nm excitation. The same procedure was applied to measure five FG-domain variants. For the variant A221-S338, we were unable to identify cells containing well-formed condensates, possibly due to low GCE efficiency at this site. As a result, we measured an alternative variant, A221-T353, instead.

Regions of interest (ROIs), corresponding to nuclear condensates, were selected based on intensity and average lifetime per pixel (Fig. 5d). Time-resolved donor fluorescence intensity profiles were extracted from ROIs before and after acceptor photobleaching. The

total fluorescence decay was computed by combining the parallel and perpendicular polarisation components ($I_\parallel$ and $I_\perp$) using[78]:

$$I(t) = (1 - 3L_2) G I_\parallel(t) + (2 - 3L_1) I_\perp(t) \qquad (9)$$

where $L_1$ and $L_2$ account for polarisation mixing caused by the high-numerical-aperture objective, and $G$ is the factor accounting for the difference in the detection efficiencies $\eta$ between parallel and perpendicular polarisation, given by Eq. (10):

$$G = \frac{\eta_\perp}{\eta_\parallel} \qquad (10)$$

### Lifetime-based analysis of FRET measurements

For cellular measurements, FLIM data were analysed according to a previously developed strategy[40,51]. The measured fluorescence signal from the donor channels comprises three populations: the donor-only population $I_{Donly}$, the FRET population between donor and acceptor dyes $I_{FRET}$, and the cellular background signal $I_{bg}$. The time-resolved fluorescence intensity in the donor channel can thus be described by Eq. (11).

$$I(t) = \left[ I_{Donly}(t) + I_{FRET}(t) + I_{bg}(t) \right] \otimes IRF$$
$$= \left\{ A_D e^{-\frac{t}{\tau_D}} + A_{FRET} \int_0^\infty \rho(r) e^{-\frac{t}{\tau_D}} \left[ 1 + \left( \frac{R_0}{r} \right)^6 \right] dr + A_{bg} \sum_{i=1}^N \alpha_i e^{-\frac{t}{\tau_{bg_i}}} \right\} \otimes IRF$$
$$(11)$$

$$A_D + A_{FRET} + A_{bg} = 1 \qquad (12)$$

where $A_D$, $A_{FRET}$, and $A_{bg}$ represent the amplitude fractions (initial intensities at $t = 0$) of the donor-only, FRET, and background components, respectively. $\tau_D$ is the donor lifetime in the absence of FRET, $r$ describes the inter-residue distance distribution based on a Gaussian chain model, and $R_0$ is the Förster distance of the dye pair. $\tau_{bgi}$ and $\alpha_i$ characterise the background decay components. The IRF is the instrument response function and was obtained using a freshly prepared saturated solution of potassiumiodide and erythrosine B[79].

Background lifetimes ($\tau_{bgi}$ and $\alpha_i$) were determined from ~40 cells expressing the amber-mutant NHA9 in the presence of Boc-lysine, a non-reactive ncAA, and fitted using a four-exponential model via TauFit in PAM[76]. After acceptor photobleaching, FRET contributions are eliminated, simplifying the signal to:

$$I'(t) = \left\{ A_D e^{-\frac{t}{\tau_D}} + A_{FRET} e^{-\frac{t}{\tau_D}} + A_{bg} \sum_{i=1}^N \alpha_i e^{-\frac{t}{\tau_{bg_i}}} \right\} \otimes IRF \qquad (13)$$

Thus, the background $A_{bg}$ and $\tau_D$ was obtained by fitting with the acceptor photobleaching sample. To isolate $A_D$, the pre-photobleaching data were fitted using Eq. (11), particularly for samples with short inter-residue distances (i.e., NHA9$^{A221TAG-S283TAG}$ in this case), where FRET signals are clearly distinguishable from the donor-only signals. Following this, the determined $\tau_D$ and $A_{bg}$, together with $R_0$, $\sum_{i=1}^N \alpha_i e^{-\frac{t}{\tau_{bg_i}}}$, $A_D$, Eq. (11) was used in conjunction with a maximum likelihood estimator (MLE) to extract the root-mean-square inter-residue distance $R_E$. For each mutant, cells with phasor values outside the mean ± 3 standard deviations were excluded from further analysis.

Bootstrap resampling was applied to each mutant to account for cellular heterogeneity. In each round, 70% of the data were randomly sampled, and the aggregated fluorescence decay profiles were globally fitted using the Gaussian chain model via MLE in PAM to determine the scaling exponent $v$. This resampling and fitting procedure was repeated 10 times. For individual mutant analysis, $R_E$ was first extracted for

each construct, followed by fitting $R_E$ versus $N_{res}$ to derive the apparent scaling exponent $v$.

For the lifetime-based analysis of smFRET measurements of nanocluster, background signal was recorded using buffer and $A_D$ was set to 0 when applying Eq. (11) to extract $R_E$. In case of lifetime-based analysis of in vitro condensates, the background signal was obtained by measuring 10 μM of unlabelled NHA9. To determine $\tau_D$ and $A_D$, 1 nM donor-only labelled NHA9 mixed with 10 μM of unlabelled NHA9 was measured. The extracted $\tau_D$ and $A_D$ was then used to obtain $R_E$. Fluorescence anisotropy data are presented in Supplementary Fig. 12a.

## Phasor plot analysis of FLIM data
The phasor plot, also known as the polar plot, provides a graphical representation of raw FLIM data in a vector space. In this method, each fluorescence decay profile is transformed into a single point on the phasor plot. This transformation is achieved by converting the time-domain fluorescence decay into the frequency domain using a Fourier transform. The phasor coordinates are computed according to the following equations:

$$g_{i,j}(\omega) = \int_0^T I(t) \cdot \cos(\omega t)\mathrm{d}t / \int_0^T I(t)\mathrm{d}t \tag{14}$$

$$s_{i,j}(\omega) = \int_0^T I(t) \cdot \sin(\omega t)\mathrm{d}t / \int_0^T I(t)\mathrm{d}t \tag{15}$$

in which $g_{i,j}(\omega)$ and $S_{i,j}(\omega)$ are the $x$ and $y$ coordinates of the phasor plot, $\omega$ is the angular frequency of excitation, and $T$ is the period of the laser pulses. To ensure accurate scaling of the phasor coordinates, the system was first calibrated by applying a Fourier transform to the measured instrument response function, which was designated as the zero-lifetime ref. 80. Each subsequent phasor point derived from the FLIM data was then calibrated using the same parameters, thereby referencing the final phasor plot to the established calibration standard. The above procedure was performed with codes in MATLAB from a previous study[40].

## Molecular Dynamics simulations
The Molecular Dynamics (MD) simulations were carried out using the Hoomd-Blue package version 5.0.1 patched for ROCm 6.3 support[81]. We used gsd 2.9.0 to save the trajectories as binary files. For non-bonded interactions, we used the HPS-Urry force field, which is based on the Urry hydrophobicity scale[57]. Bonded parameters were defined as 3.8 Å and 10 kJ/Å² respectively[82]. We used a cutoff radius of 20.0 Å for the ashbaugh-hatch interactions and 35.0 Å for the yukawa potential. The alpha-helices and tertiary structure motif in HOXA9 were kept rigid using elastic networks composed of harmonic bonds with the same bond rigidity[83]. Bonds were created between atoms within less than 12.0 Å from each other. The coordinates for the structured regions of HOXA9 are alpha-fold predictions taken from UniProt database (https://www.uniprot.org/uniprotkb/P31269/entry)[84,85]. For the simulation of NHA9 with the GLEBS domain, coordinates for secondary and tertiary structure of the GLEBS domain were taken from PDB:3MMY, and the same elastic network procedure used for HOXA9 was used to constrain its structure. HPS-Urry was parameterised for IDPs, therefore introducing the elastic network to keep the structure of the DBD rigid may increase the inaccuracy of the simulation results. Previous works have, nonetheless, used HPS-Urry and other similar force fields to simulate partially structured sequences. On top of elastic networks, rigid body constraints have also been used in other works to simulate IDPs with folded regions. Both approaches have been shown to yield qualitatively correct results for chain conformations and phase separation propensities[58–61].

All simulations were carried out at 300 K and temperature was kept constant using the Langevin thermostat, with a drag coefficient of 10.0 $m_u$/ps ($m_u$ being the atomic mass unit). We created cubic boxes with dimensions chosen in order to maintain a fixed volume fraction of 0.001 (details in Supplementary Table 3). The timestep was set to 10 fs. Trajectories were output at intervals of 1 ns.

Individual protein chains were initialised in proximity to each other with random conformations, except for those in the HOXA9 block. We then performed an equilibration step consisting of a 100 ns interval where non-bonded interactions are progressively turned on and the timestep was reduced to 0.01 fs and temperature to 150 K, followed by a 24 ns interval where temperature was gradually increased to 300 K, followed by a 56 ns interval where the time step was gradually increased to 10 fs. Throughout all these steps a spherical wall was used to confine the protein chains in order to induce phase separation quicker. We then removed the walls and ran the simulations for 820 ns to further relax the system. In some cases, we observed unphysical entanglements between beads in the disordered parts with helices in the DBDs. In those cases, we reinitialised the simulation in the same way but with chain conformations extracted from the single chain simulations and kept non-bonded interactions at full strength to avoid such entanglements.

Single chain simulations were set up in a different manner by placing 64 copies in a lattice grid at very large distances from each other in a very large cubic box of side 2400 Å. To ensure that no interaction occurred between the individual chains we fixed the position of the first bead in each chain in space. By following this setup we were able to generate 64 independent single chain trajectories using less computational resources than we would have otherwise used running 64 independent single chain simulations. The equilibration step for single chain simulations was identical to the multi chain one except that no spherical walls were used. We also observed unphysical entanglements in 9 of the 64 chains, and those were ignored in the data analysis, the sample size was still 53 which was more than enough to obtain good statistics.

Our production run had a length of at least 7 μs, on which we carried out our analysis. The number of frames per simulation is given in the last column of Supplementary Table 3. The simulation for NHA9 including the GLEBS domain had a production run of 2.6 μs. NHA9 showed a remarkably high phase separation propensity in our simulations, very few chains were observed in the dilute phase in any of the simulations (Supplementary Table 3). This was confirmed by using the clustering algorithm from the freud python package (v2.13.2) based on amino acid positions where 2 chains containing at least one pair of amino acids within a distance of less than 7.0 Å were considered to be in the same cluster[86]. The apparent Flory exponent ($v$) was computed by fitting Eq. 6 to the data in Fig. 6a, with the error bars taken as the square root of the diagonal term of the covariance matrix of the fit corresponding to $v$. Error bars in Fig. 6a, were computed as the standard error on the mean (SEM) of each pair distance: $SEM = SD/\sqrt{n_{eff}n_{chains}}$. The quantity $n_{eff}$ is the effective number of frames, and corresponds to the number of statistically uncorrelated frames for each pair of amino acids. It is defined as $n_{eff} = n_{frames}/\tau$, where $n_{frames}$ is the total number of frames and $\tau$ is the relaxation time extracted from the autocorrelation function of distances of each pair of amino acids by fitting an extended exponential function. The quantity $n_{chains}$ corresponds to the number of chains in the NHA9 cluster and $SD$ is the standard deviation. The values computed for $n_{eff} \cdot n_{chains}$ are summarized in Supplementary Table 4. For NHA9 simulations without the GLEBS domain, between 8965 and 454,336 statistically independent FRET distance values were obtained, resulting in a very low SEM. These results confirm that the simulations were run for sufficiently long durations to ensure proper equilibration of FRET distances and to eliminate dependence on initial conditions. For NHA9 simulations including the GLEBS domain, between 2,310 and 26,900 statistically independent FRET distance values were obtained.

To compute the density profiles in Fig. 6d and Supplementary Figs. 15–18 and 20 we first computed the position of the centre of mass (COM) of the cluster at each frame. We then computed histograms of the distance from the COM for each of the $n_{aa}^{chain}$ amino acids in the sequence (524 for NHA9 without GLEBS domain and 581 for NHA9 with GLEBS domain). In order to compute the density profiles of each individual block we summed the histograms corresponding to each of the constituent $n_{aa}^{block}$ amino acids in the block. We then multiplied it by the factor $n_{aa}^{chain}/n_{aa}^{block}$, which accounts for the different lengths of the blocks compared to the whole chain. We then obtain a density in the unit of moles of amino acids per liter. Error bars were computed as the SEM of the density at each bin of the density profile of each analyzed block. The same formula used in the calculation of the error bars of the FRET distances was used. However, in this case, we measure the relaxation time $\tau$ to be lower than one frame for every bin of every density profile obtained, so $n_{eff} = n_{frames}$.

To compute contact maps in Supplementary Fig. 19, we considered amino acids to be neighbours if they were within a distance of $1.5\sigma$ from each other, where sigma was taken as the mean sigma of the amino acid pair in the Ashbaugh-Hatch term of the force field, $\sigma = 1/2(\sigma_1 + \sigma_2)$. We calculated the interchain contacts for every pair of NHA9 amino acids within the cluster and then divided them by the mean number of chains times the number of frames to obtain units of probability of contact per chain.

### Sequence property profile of NHA9

Our plots of the sequence hydrophobicity, fraction of charged residues (FCR), and net charge were based on the HPS-Urry force field. We defined the hydrophobicity of each amino acid as the $\lambda$ parameter. We used a sliding window of size 25 amino acids to smooth the data. We defined the x-coordinate as the centre of the window and omitted the first 12 and last 13 amino acids from the plot.

### Reporting summary

Further information on research design is available in the Nature Portfolio Reporting Summary linked to this article.

## Data availability

All trajectories from MD simulations and a corresponding data analysis of the contact maps corresponding to Supplementary Fig. 19 are deposited on the Edmond repository[87] (https://doi.org/10.17617/3.NIKDDY). All other data are available in the main text or Supplementary Information. Source data are provided with this paper.

## Code availability

Codes related to simulations and analysis are available at https://gitlab.mpcdf.mpg.de/dillenburgr/nha9.

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

## Acknowledgements
We acknowledge funding by the Deutsche Forschungsgemeinschaft (DFG, German Research Foundation)–SFB1551–"Polymer concepts in Cellular Function" Project No. 464588647, as well as its internal service project on Biopolymer Engineering and Bioanalytics (IMB protein production core facility), especially Martin Möckel and at the MPI core facility, especially Christine Rosenauer, Ute Heinz and Svenja Morsbach. We thank Nadje Hellmann and Dirk Schneider for their help with CD spectra data acquisition. E.A.L. also acknowledges funding from the ERC-ADG grant 'MultiOrganelleDesign'. This work was supported by the Max-Planck Computing and Data Facilities.

## Author contributions
H.R. M.G. S.W. and E.A.L. designed the conceptual framework of the study. H.R. and E.H. performed the experiments. R.F.D. conducted molecular dynamics simulations. H.R., R.F.D., E.H., S.W., M.G. and E.A.L. contributed to data acquisition and interpretation. H.R., R.F.D., S.W., M.G. and E.A.L. wrote the manuscript. M.G. and E.A.L conceived the work.

## Funding

## Competing interests
E.A.L. holds several patents related to genetic code expansion and is a cofounder of Veraxa GmbH, a company specialising in the generation of antibody-drug conjugates via GCE. The remaining authors declare no competing interests.
