## [Transparent Peer Review file · Nature Communications]

Differential conformational expansion of NUP98-HOXA9 oncoprotein from nanosized assemblies to macrophases

Corresponding Author: Professor Edward Lemke

Version 0:

Reviewer comments:

Reviewer #1

(Remarks to the Author)

Summary of the manuscript

Ruan et al. combine both experiments and simulations to explore the different conformational state of a chimeric transcription factor protein Nup98-HOXA9 (NHA9), which is implicated in leukemogenesis. More specifically, this protein has an FG-enriched intrinsically disordered region and a folded domain. The authors used a series of single molecule characterization experiments and simulations to show how the compactness changes from dilute solution to phase separated condensate. The apparent Flory scaling exponents obtained experimentally and simulationally are consistent with each other. From simulations, the authors have found that oligomeric state of NHA9 is structurally similar to micelle, and this might have biological implication.

Comments

With the condensate field evolving, it is becoming more and more important to get an understanding of the biomolecular conformations at a molecular level. The detailed experiments and simulations in this manuscript provides such type of information, thus it helps researchers get better understanding of the biological implications of biomolecules during condensation processes. However, I have some comments regarding the manuscript.

1. In describing Figure 2a about the heterogenous cluster formation (Page 5), the author described as "As the concentration of unlabeled NHA9 increased, the larger nanoclusters with heterogeneous size distributions progressively emerged." However, what I see from the figure is that from 10nM to 2uM, the distribution of peaks around to $<50 \text{ um}^2/\text{sec}$ (I assume this range corresponds to larger clusters) actually disappears, and only two major peaks appear in 2uM concentration. Can authors explain how to interpret the emergence of heterogeneous size distributions with increased concentrations?

2. For single chain simulations, the authors used an unusual approach instead to simulate many sufficiently separated chains by fixing the first bead of the chain in space. However, by doing so, each single chain's degrees of freedom is less than a freely-floating single chain, and thus there is possibility to introduce some bias to the conformations. The authors may want to be careful about this approach. Therefore, I suggest authors to perform additional single chain simulations without fixing the first bead and compare the results to see if the Flory scaling exponents are consistent.

3. Due to the computational limit, the authors were not able to simulate larger cluster size (> 50) to see if the Flory scaling exponents are consistent with in vitro experiments. From results, the authors speculated further increases in cluster size would lead to further expansion of the IDP block, and increase the observed value of Flory scaling exponent. This means that the surface effect of nanoclusters decreases. In the actual macro phase separation, there is no surface

effect to the molecules. In simulation, there is a way to probe this approximately with reduced computational cost. In NPT ensemble, the authors can prepare a system with a little more chains in a periodic box, slowly compress to form a dense phase and relax at zero pressure. In such case, there will be no interface, and the simulation is basically sampling a dense phase. By measuring the chain conformations in this ensemble, the authors should be able to get some idea for chain conformations in dense phase.

4. The authors used an elastic network model for the folded domain in coarse-grained simulations, and observed the micelle structure in the nanoclusters. As the authors pointed out in the manuscript, HPS-Urry force field is mainly developed for intrinsically disordered proteins, not for the folded domains. The elastic network is a more coarse-grained (crude) approximation in this case, and overlooks a lot of specific folded-domain interactions in such representations. Therefore, the authors may want to be careful about interpreting the results regarding the behaviors obtained for the folded domain, and make the approximations and limitations clearer in the main text.

5. In page 3, the authors mentioned "The disordered FG domain of NHA9 plays essential roles in activating oncogenic gene expression". This doesn't seem to be a common knowledge, I wonder if the authors could add some supporting references for this argument.

6. In Figure 1 d, the authors should also include labeled residues for each 2d histogram, as they did for Fig 1c.

7. In Figure 6d, the authors should add error bars or at least standard deviation of the density profile to be consistent.

Reviewer #2

(Remarks to the Author)

Summary of key results

This paper from Prof. Lemke's lab provides a detailed biophysical characterization of the oncofusion protein Nup98-HOXA9 (NHA9), in which an intrinsically disordered FG repeat region of Nup98 is fused to the DNA-binding domain of the transcription factor HOXA9. The authors employ a variety of approaches, including site-specific labeling in vitro and in cells, single-molecule FRET, FLIM-FRET, FCS, and MD simulations. Using these tools, the authors find that NHA9 displays a relatively compact/collapsed configuration in the dilute state but gradually expands in a concentration-dependent manner as the protein forms nanoclusters and macroscopic condensates in the sub-saturated and saturated regimes, respectively. The polymer scaling exponents within condensates are in close agreement between in vitro and in cell measurements. Chain expansion is likely due to a gradual trading of intramolecular interactions in the dilute state for intermolecular interactions in the oligomeric or condensed states. Simulations suggest that the nanoclusters display properties of weak micelles, with the DNA-binding domain enriched at the condensate periphery. The authors propose that concentration-dependent chain expansion and micelle-like properties of clusters may facilitate interactions with both DNA and components of the transcriptional machinery. Overall, this work provides important, mechanistic insight into the biophysical properties of IDP-containing transcription factors and oncofusions, with implications for our understanding of how they drive transcription in health and disease. The data is thorough, convincing, and well-presented. The cell experiments involving genetic code expansion and orthogonally translating organelles for site-specific labeling are particularly impressive. This reviewer strongly recommends publication after addressing a few points, described below.

Main points

1. The primary approach used to characterize the IDP scaling behavior of NHA9 involves site-specific labeling for FRET, with a donor at a fixed position and an acceptor at varying positions of increasing distance from the donor. Thus, each FRET pair effectively interrogates the behavior of a segment of the chain between the two fluorophores. Can the authors comment on how each of the five acceptor positions were chosen? Were they picked at random? If not random, were the positions chosen such that the average properties of the chain segments between the fluorophores were roughly similar in terms of amino acid composition, net charge per residue, average hydrophilicity, etc.? This may be an important point to clarify, as the analysis of RE vs. Nres (e.g. in Fig. 1e) likely requires that the chain segments are chemically similar to each other. However, some IDPs can display blocky characteristics, possibly causing different sub-segments to have different average compositions. While this may not be a concern for NHA9, as the model appears to fit the data well, it might be helpful to include some discussion on the limitations of this approach as a general tool for interrogating the scaling behavior of other IDPs.

2. In the first paragraph of the Results section, the authors state that "NHA9 resembles a diblock copolymer, in which the disordered FG domain functions as a hydrophobic block, while the structured DBD serves as a hydrophilic block." While data presented later in the paper support this idea, the only piece of information provided at this point is the disorder plot in Fig. 1b. Is it possible to draw conclusions regarding relative hydrophobicity along a protein sequence from a plot of disorder prediction alone? If not, please provide additional rationale to support this statement at this early point in the paper, or consider revising to avoid potential confusion.

3. The suppressive effect of DNA on NHA9 nanocluster formation seems rather non-intuitive. A naïve assumption would be that DNA should promote nanocluster formation, as the additional, heterotypic binding interaction between NHA9 and DNA could conceivably strengthen the driving forces for oligomerization and condensation. The authors' interpretation of this suppressive effect seems to be that NHA9 exchanges self-interactions for DNA interactions, solubilizing the oligomers, but it is unclear why this might be. Is this a charge-driven effect, in which electrostatic interactions between NHA9 and DNA are

stronger than the hydrophobic interactions that drive oligomerization? More discussion on the possible mechanism of this effect would be helpful.

4. Can the authors analyze the gel shift data in Supplementary Fig. 1b, or perform additional gel shift experiments, to estimate the affinity, K_D , of NHA9 for the DNA fragment? How does this K_D value compare to the concentration regimes at which nanoclusters and condensates form, and could the authors interpret how DNA might shift these concentration regimes? Understood that the “compensatory” effects of the large DNA molecules make it challenging to analyze the loss of nanoclusters at lower NHA9 concentrations, but a bit of discussion or speculation on this point might be helpful.

5. Each 2D histogram in Fig. 1d appears to correspond to the same donor-acceptor pairs indicated in Fig. 1c, but this is not immediately obvious. Consider adding labels to each plot in Fig. 1d.

Reviewer #3

(Remarks to the Author)

NHA9 is a fusion oncoprotein driving acute myeloid leukemia, myelodysplastic syndrome, and chronic myeloid leukemia. Ruan et al. characterized the conformational changes of NHA9 using smFRET, FLIM-FRET, coarse-grained simulations, and phase separation assays in their monomeric, oligomeric, and densely packed condensate states. They found that NHA9 behaves as a weak amphiphile, with continuous expansion of the FG domain during the transition from dilute to oligomeric state. With simulations, they found that NHA9 demonstrated a non-random and non-core-shell spherical micelle-like organization at the oligomeric state. The experiments are well-designed and carefully validated, and their findings offered insights into the dynamic nature of NHA9 during condensate formation.

Major points to address:

1. The schematic in Fig. 1a is confusing and does not show the removal of the GLEBS domain.
2. The GLEBS domain seems to be preceding the residues probed during the in vitro experiments. However, whether the overexpression assay in live cells and simulation uses full-length NHA9 or Δ GLEBS NHA9 is unclear. Would live cell expressing Δ GLEBS NHA9 show the same result as full-length NHA9? Similarly, would simulating Δ GLEBS NHA9 show the same result as full-length NHA9?
3. NHA9 was purified under denaturing conditions. Can the authors provide further validation that the protein refolds during in vitro experiments?
4. In the FCS data, it is unclear how the diffusion coefficient translate to cluster size. The authors made conclusions on nanocluster sizes but the data shown are all in diffusion coefficients (for example, Figs 2a, 3a).
5. In addition, the 3-axis plot for FCS data is not helpful for visualizing the relative sizes of different peaks. May as well display them in 2-axis and separate to different concentrations similar to FRET data.
6. It will be preferred to include a negative control for Supp Fig. 1b showing NHA9 doesn't bind to a scrambled HOXA9 binding site.
7. The authors only measured the scaling component of NHA9 in the droplet phase in cell. How about the scaling component in the diffuse phase in cell?
8. The authors showed representative images for labeling in cell (Figs. 5b,c). It is ideal to quantify the colocalizations across many different cells. Fig.5c can be compared to non-colocalization to heterochromatin marks such as H3K9me3.

Minor points

1. Human proteins/genes should be upper case (e.g., NUP98).
2. Line 65, “NHA9 condensates has” should be “have”.
3. Line 231 should be Fig.2e not 4c. Also check and confirm other figure labelings are correct.
4. Line 318, “...can be explain” should be “...can be explained”.
5. Line 418 “blockcopolymer” should be two words.
6. Fig. 6c: the authors concluded that the DNA-binding regions are mostly outside the cluster. But without showing the inside of the cluster, the readers are not convinced DNA-binding regions are not inside. It will be preferably to draw the cluster transparently to show the inside.

Reviewer #4

(Remarks to the Author)

Version 1:

Reviewer comments:

Reviewer #1

(Remarks to the Author)

The authors have addressed the points that I have raised, and I am glad to see the quality of the paper have been improved by the work done by authors for the revision. Therefore, I recommend for publication.

(Remarks on code availability)

The repo contains simulation codes and analysis codes with README files including sufficient descriptions. I was able to download and visualize the example outputs.

Reviewer #2

(Remarks to the Author)

I would like to sincerely thank the authors for taking the time to carefully respond to each of my points. The authors have done a commendable job, and I believe their revisions have strengthened the paper. I have no further comments or requests, and I look forward to seeing this exciting work published in Nature Communications.

(Remarks on code availability)

Reviewer #3

(Remarks to the Author)

The revised manuscript is much improved. I don't have further comments.

(Remarks on code availability)

N/A

Reviewer #4

(Remarks to the Author)

(Remarks on code availability)

Point-by-point

Reviewer comments in black

Our response in blue

Changes in main text of the manuscript and in the Supplementary information were highlighted in yellow.

Reviewer #1 (Remarks to the Author):

Summary of the manuscript

Ruan et al. combine both experiments and simulations to explore the different conformational state of a chimeric transcription factor protein Nup98-HOXA9 (NHA9), which is implicated in leukemogenesis. More specifically, this protein has an FG-enriched intrinsically disordered region and a folded domain. The authors used a series of single molecule characterization experiments and simulations to show how the compactness changes from dilute solution to phase separated condensate. The apparent Flory scaling exponents obtained experimentally and simulationally are consistent with each other. From simulations, the authors have found that oligomeric state of NHA9 is structurally similar to micelle, and this might have biological implication.

Comments

With the condensate field evolving, it is becoming more and more important to get an understanding of the biomolecular conformations at a molecular level. The detailed experiments and simulations in this manuscript provides such type of information, thus it helps researchers get better understanding of the biological implications of biomolecules during condensation processes. However, I have some comments regarding the manuscript.

We thank the reviewer for the positive and thorough review of our work and for the insightful comments!

1. In describing Figure 2a about the heterogenous cluster formation (Page 5), the author described as "As the concentration of unlabeled NHA9 increased, the larger nanoclusters with heterogeneous size distributions progressively emerged." However, what I see from the figure is that from 10nM to 2uM, the distribution of peaks around to $<50 \mu\text{m}^2/\text{sec}$ (I assume this range corresponds to larger clusters) actually disappears, and only two major peaks appear in 2uM concentration. Can authors explain how to interpret the emergence of heterogeneous size distributions with increased concentrations?

We apologize for the confusion. In Figure 2a, the diffusion coefficients of the particles are presented. The short sampling time FCS method was specifically chosen as it allows one to get an idea what particles exist in solution, despite being heterogenous. The hydrodynamic radius of the particles was estimated using the Stokes–Einstein equation $D = \frac{k_B T}{6\pi\eta r}$, where D is the translational diffusion coefficient, k_B is Boltzmann's constant, T is the absolute temperature, η is the solvent viscosity, and r is the hydrodynamic radius of the particle. Based on this

relationship, diffusion coefficients of $70 \mu\text{m}^2/\text{s}$, $50 \mu\text{m}^2/\text{s}$, $10 \mu\text{m}^2/\text{s}$ correspond to hydrodynamic radii of 3.2 nm, 4.5 nm and 22.7 nm, respectively. The distribution of diffusion coefficients with peaks below $10 \mu\text{m}^2/\text{s}$ reflects the presence of large clusters. As the concentration of unlabelled NHA9 increased, this peak progressively increased, indicating that larger clusters kept on forming. We now improved visualization of the data, by replotting the diffusion coefficient distributions shown in Figures 2a and 3a, converting them from a two-dimensional peak display to a three-dimensional peak representation.

2. For single chain simulations, the authors used an unusual approach instead to simulate many sufficiently separated chains by fixing the first bead of the chain in space. However, by doing so, each single chain's degrees of freedom is less than a freely-floating single chain, and thus there is possibility to introduce some bias to the conformations. The authors may want to be careful about this approach. Therefore, I suggest authors to perform additional single chain simulations without fixing the first bead and compare the results to see if the Flory scaling exponents are consistent.

The reviewer's concern is justifiable and we were also faced with this question when implementing the aforementioned setup for single chain simulations. If we consider the Rouse model of polymer dynamics, the "freezing" of the coordinates of the initial - or any other individual - bead of the protein chain is equivalent to a translation operation. We effectively suppress the 0th mode of the chain; its partition function, however, is unaffected by translation operations. That means chains' conformations are not affected, and we can therefore expect the R_g or R_e (end-to-end distance) of our chains to yield identical values in both simulation setups. In order to confirm such statement, we performed a single chain simulation in a very large simulation box containing only one chain where none of the beads were fixed in space. Our results show that we obtain identical distributions for both R_g and R_e (see the figure below).

Comparison of R_g (left) and R_e (right) distributions for both methods. Black curves represent one individual chain free to explore the whole simulation box and red curves represent a system with multiple individual chains spaced far apart from each other, whose first bead is unable to move, preventing interchain interaction. The distributions are nearly identical, with the simulation containing only one chain showing more noise due to less sampling.

3. Due to the computational limit, the authors were not able to simulate larger cluster size (> 50) to see if the Flory scaling exponents are consistent with in vitro experiments. From results, the authors speculated further increases in cluster size would lead to further expansion of the IDP block, and increase the observed value of Flory scaling exponent. This means that the surface effect of nanoclusters decreases. In the actual macro phase separation, there is no surface effect to the molecules. In simulation, there is a way to probe this approximately with reduced computational cost. In NPT ensemble, the authors can prepare a system with a little more chains in a periodic box, slowly compress to form a dense phase and relax at zero pressure. In such case, there will be no interface, and the simulation is basically sampling a dense phase. By measuring the chain conformations in this ensemble, the authors should be able to get some idea for chain conformations in dense phase.

We thank the reviewer for the suggestion. The reviewer is correct that such simulations would provide additional insight into chain conformations. They would be, however, too costly from the computational side. Equilibrating polymer melts can be very costly, especially for very long chains like NHA9, which exhibits a high degree of entanglement. The reason we were able to equilibrate droplets of size up to 50 chains is due to the presence of the interface. The interface has lower density, which has faster relaxation times. By removing the interface, equilibration times would go up significantly, on top of that we would also need more than 50 chains in the box to perform the NPT simulations which would make it even slower. This is because we would need a box with sides larger than the maximum end-to-end distance of the chains, to avoid chains from interacting with themselves due to the periodic boundary conditions.

It would be interesting to see whether conformations of chains in a melt phase undergoes any type of microstructure or microphase separation. It is however particularly challenging in the case of NUP98/HOXA9; other IDP/TF fusions may provide a better system to investigate this, although experimentally challenging. Nevertheless, our inability to determine dense phase conformation does not alter the conclusions of the current article.

4. The authors used an elastic network model for the folded domain in coarse-grained simulations, and observed the micelle structure in the nanoclusters. As the authors pointed out in the manuscript, HPS-Urry force field is mainly developed for intrinsically disordered proteins, not for the folded domains. The elastic network is a more coarse-grained (crude) approximation in this case, and overlooks a lot of specific folded-domain interactions in such representations. Therefore, the authors may want to be careful about interpreting the results regarding the behaviors obtained for the folded domain, and make the approximations and limitations clearer in the main text.

The reviewer is correct in their statement. This approach, while not the most precise, has shown to produce qualitatively correct results in previous studies (refs. 58-61 in the maintext). We added the sentences below to the manuscript in order to highlight the potential shortcomings of our simulation setup.

“Even though the HPS-Urry force field and the use of the elastic network do not correctly capture side-chain geometry and folded-domain interactions, it has nonetheless been shown to provide qualitatively reliable results in IDPs self assembly.” - in section: Molecular simulations confirm NHA9's IDP expansion within nanocluster and reveal micelle-like structure.

“HPS-Urry was parameterized for IDPs, therefore introducing the elastic network to keep the structure of the DBD rigid may increase the inaccuracy of the simulation results. Previous works have, nonetheless, used HPS-Urry and other similar force fields to simulate partially structured sequences. On top of elastic networks rigid body constraints have also been used in other works to simulate IDPs with folded regions. Both approaches have shown to yield qualitatively correct results for chain conformations and phase separation propensities.” - in methods section

5. In page 3, the authors mentioned "The disordered FG domain of NHA9 plays essential roles in activating oncogenic gene expression". This doesn't seem to be a common knowledge, I wonder if the authors could add some supporting references for this argument.

We thank the reviewer for pointing this out. We have rephrased this sentence to: “Fusion of the disordered FG domain of NHA9 is a known genetic phenotype for a specific form of acute myeloid leukemia (Ahn, J. H. *et al.*, *Nature* **595**, 591–595, 2021). We have cited this paper in the revised main text.

6. In Figure 1 d, the authors should also include labeled residues for each 2d histogram, as they did for Fig 1c.

We thank the reviewer for the suggestion and we revised accordingly.

7. In Figure 6d, the authors should add error bars or at least standard deviation of the density profile to be consistent.

We thank the reviewer for pointing this out. In response, we have added error bars to the plot and included a detailed description of their calculation in the Methods section.

Reviewer #2 (Remarks to the Author):

Summary of key results

This paper from Prof. Lemke's lab provides a detailed biophysical characterization of the oncofusin protein Nup98-HOXA9 (NHA9), in which an intrinsically disordered FG repeat region of Nup98 is fused to the DNA-binding domain of the transcription factor HOXA9. The authors employ a variety of approaches, including site-specific labeling in vitro and in cells, single-molecule FRET, FLIM-FRET, FCS, and MD simulations. Using these tools, the authors find that NHA9 displays a relatively compact/collapsed configuration in the dilute state but gradually expands in a concentration-dependent manner as the protein forms nanoclusters and macroscopic condensates in the sub-saturated and saturated regimes, respectively. The polymer scaling exponents within condensates are in close agreement between in vitro and in cell measurements. Chain expansion is likely due to a gradual trading of intramolecular interactions in the dilute state for intermolecular interactions in the oligomeric or condensed states. Simulations suggest that the nanoclusters display properties of weak micelles, with the DNA-binding domain enriched at the condensate periphery. The authors propose that concentration-dependent chain expansion and micelle-like properties of clusters may facilitate interactions with both DNA and components of the transcriptional machinery. Overall, this work provides important, mechanistic insight into the biophysical properties of IDP-containing transcription factors and oncofusins, with implications for our understanding of how they drive transcription in health and disease. The data is thorough, convincing, and well-presented. The cell experiments involving genetic code expansion and orthogonally translating organelles for site-specific labeling are particularly impressive. This reviewer strongly recommends publication after addressing a few points, described below.

We thank the reviewer for the positive and thorough review of our work and for the insightful comments!

Main points

1. The primary approach used to characterize the IDP scaling behavior of NHA9 involves site-specific labeling for FRET, with a donor at a fixed position and an acceptor at varying positions of increasing distance from the donor. Thus, each FRET pair effectively interrogates the behavior of a segment of the chain between the two fluorophores. Can the authors comment on how each of the five acceptor positions were chosen? Were they picked at random? If not random, were the positions chosen such that the average properties of the chain segments between the fluorophores were roughly similar in terms of amino acid composition, net charge per residue, average hydrophilicity, etc.? This may be an important point to clarify, as the analysis of RE vs. Nres (e.g. in Fig. 1e) likely requires that the chain segments are chemically similar to each other. However, some IDPs can display blocky characteristics, possibly causing different sub-segments to have different average compositions. While this may not be a concern for NHA9, as the model appears to fit the data well, it might be helpful to include some discussion on the limitations of this approach as a general tool for interrogating the scaling behavior of other IDPs.

We thank the reviewer for the helpful suggestions to include a discussion on labelling position choice and the scaling behavior of IDPs. The N-terminal region of NHA9, derived from a component of the nuclear pore complex, contains two IDPs enriched in phenylalanine-glycine (FG) motifs, separated by a GELBS domain responsible for binding to the RAE1 protein (Fig. 1a). In this study, we focus on the second FG domain, which is highly hydrophobic, displays a high density of FG motifs, and is strongly depleted of charged residues.

As we want to experimentally test how this FG domain behaves, we designed more than one FRET pair to sample that region. We selected five labelling sites located in inter-FG spacers approximately every 30 residues (labelling sites are highlighted in red and FG motifs in green in the following textbox; also see in the revised supplementary Table 2). In choosing these sites, we aimed to avoid phenylalanine, glycine, charged residues, and highly hydrophobic residues. Based on these considerations, we selected the five labelling sites. We have added further explanation in the revised maintext.

As noted above, the FG domain of NHA9 contains numerous FG motifs. To further support our labelling choice, we reanalyzed the NHA9 sequence and plotted hydrophobicity and fraction of charged residues (see revised Fig. 1c). Both hydrophobicity and charge distribution appear homogeneous across the FG domain. Thus, scaling behavior derived from our selected positions should be representative of the overall properties of the second FG domain.

The reviewer correctly pointed out that some IDPs can display blocky characteristics. We agree. In such cases, scaling behavior should indeed be probed separately for each segment. We have added a discussion of this scenario in the revised Discussion section.

```
SGSMFNKSFSGTFFGGGTGGFGTTSTFGQNTGFGTTSSGGAFGTSAFGSSNNTGGLFGNSQTKPGGLFGTSSFSQP  
ATSTSTGFGFGTSTGTANTLFGTASTGTSLFSSQNNFAQNKPTGFGNFGTSTSSGGLFGTTNTTSPFGSTSGSLF  
GPSSFTAAGPQNQVG(A221)GTTTGLFGSSPATSSATGLFSSSTTNSGFAYGQNKTAFGTSTTGFGTNPGGFLGQ  
QNQQTTS(S283)LFSKPFQATTQNTGFSFGNTSTIGQPS(S312)TNTMGLFGVTQASQPGGLFGTATNTS(S33  
8)TGTAFGTGTGLFGQNTGFGAVGS(S362)TLFGNNKLTTFSSSTTSAPSGTSSGGLFGFGTNTS(S398)GNSIFG  
SKPAPGTLGTLGAGFGTALGAGQASLFGNNQPKIGGPLGTGAFGAPGFNTTATLFGAPQAPVVDREKQPSE  
GAFSENNAENESGGDKPPIDPNNPAANWLHARSTRKKRAPYTKHQLELEKEFLFNMYLTRDRRYEVARLLNLTE  
RQVKIWFQNRMMKMKKINKDRAKDE
```

2. In the first paragraph of the Results section, the authors state that “NHA9 resembles a diblock copolymer, in which the disordered FG domain functions as a hydrophobic block, while the structured DBD serves as a hydrophilic block.” While data presented later in the paper support this idea, the only piece of information provided at this point is the disorder plot in Fig. 1b. Is it possible to draw conclusions regarding relative hydrophobicity along a protein sequence from a plot of disorder prediction alone? If not, please provide additional rationale to support this statement at this early point in the paper, or consider revising to avoid potential confusion.

We thank the reviewer for this valuable suggestion. In response, we performed sequence analysis of NHA9 and plotted the sequence property profiles (see revised Fig. 1c). As shown in Fig. 1c, the FG domain is predominantly hydrophobic and depleted of charged residues,

thereby forming a hydrophobic block. In fact, the FG domain of NUP98 is one of the most hydrophobic disordered domains that we have ever handled experimentally. In contrast, the structured DBD is less hydrophobic and enriched in charged residues, thereby constituting a charged block. Taken together, these features indicate that NHA9 resembles a diblock copolymer.

3. The suppressive effect of DNA on NHA9 nanocluster formation seems rather non-intuitive. A naïve assumption would be that DNA should promote nanocluster formation, as the additional, heterotypic binding interaction between NHA9 and DNA could conceivably strengthen the driving forces for oligomerization and condensation. The authors' interpretation of this suppressive effect seems to be that NHA9 exchanges self-interactions for DNA interactions, solubilizing the oligomers, but it is unclear why this might be. Is this a charge-driven effect, in which electrostatic interactions between NHA9 and DNA are stronger than the hydrophobic interactions that drive oligomerization? More discussion on the possible mechanism of this effect would be helpful.

We thank the reviewer for this valuable suggestion. DNA is only expected to enhance condensation if the DNA itself is of sufficient length to bring several NHA9 molecules together and thus reduce the effective concentration required for sufficient NHA9 intermolecular interactions to drive condensation. In our smFRET experiments, we employed short DNA fragments of 20 nucleotides. As we are using short DNA oligonucleotides to which only ~1 NHA9 molecules are able to bind, the DNA is not able to promote nanocluster formation in our experimental setup and rather leads to a suppression as discussed above. To further investigate the driving force of NHA9 nanocluster formation, we performed smFRET measurements in the presence of 1 μ M unlabelled NHA9 protein in buffer supplemented with 5% or 10% 1,6-hexanediol (to suppress hydrophobic interactions). In native buffer half of the NHA9 population is in heterogenous nanoclusters. This fraction reduces, to 25% in buffer containing 5% 1,6-hexanediol, and further reduces to an undetectable level in buffer containing 10% 1,6-hexanediol (see revised Supplementary Fig. 7). These results confirm that NHA9 nanocluster formation is primarily driven by hydrophobic interactions. Upon binding to DNA, the NHA9–DNA complex exhibited greater solubility compared with NHA9 alone. Furthermore, electrostatic repulsion between DNA molecules within a nanocluster may further enhance the solubilization of NHA9.

4. Can the authors analyze the gel shift data in Supplementary Fig. 1b, or perform additional gel shift experiments, to estimate the affinity, K_D , of NHA9 for the DNA fragment? How does this K_D value compare to the concentration regimes at which nanoclusters and condensates form, and could the authors interpret how DNA might shift these concentration regimes? Understood that the “compensatory” effects of the large DNA molecules make it challenging to analyze the loss of nanoclusters at lower NHA9 concentrations, but a bit of discussion or speculation on this point might be helpful.

We thank the reviewer for the suggestion. As the NHA9 clusters are heterogenous, we do not expect that a single sharp K_D can be obtained with any method. However, because we agree with the reviewer that having better knowledge on the K_D range would strengthen the work, we performed the following experiments:

1. We repeated the electrophoretic mobility shift assay (EMSA) with extended concentration titrations performed in triplicate, and fitted the band intensities to determine the binding affinity (K_D) between NHA9 and the DNA fragment. We obtained a K_D of 0.3 μ M (see revised Supplementary Fig. 2). It is important to note that because NHA9 forms nanoclusters even at concentrations as low as 10 nM, the K_D reported here should be considered an apparent K_D . This value reflects a composite parameter that accounts for both monomeric NHA9 and nanocluster NHA9 binding to DNA.
2. We performed EMSA using a shuffled DNA fragment as control. These results indicate that NHA9 binds more strongly to the DNA fragment containing the HOXA9 binding site but shows weaker binding to the scrambled sequence, consistent with our smFRET measurements (see Supplementary Fig. 6). However, because of the strong tendency of NHA9 to form clusters at higher concentrations, we were not able to go to sufficiently high concentrations to determine the apparent K_D in this case, but it is clearly much higher in number for K_D (i.e. lower affinity).

As suggested by the reviewer, we added a new discussion to page 6 and 7 with respect to these points.

5. Each 2D histogram in Fig. 1d appears to correspond to the same donor-acceptor pairs indicated in Fig. 1c, but this is not immediately obvious. Consider adding labels to each plot in Fig. 1d.

We thank the reviewer for the suggestion and we revised accordingly.

Reviewer #3 (Remarks to the Author):

NHA9 is a fusion oncoprotein driving acute myeloid leukemia, myelodysplastic syndrome, and chronic myeloid leukemia. Ruan et al. characterized the conformational changes of NHA using smFRET, FLIM-FRET, coarse-grained simulations, and phase separation assays in their monomeric, oligomeric, and densely packed condensate states. They found that NHA9 behaves as a weak amphiphile, with continuous expansion of the FG domain during the transition from dilute to oligomeric state. With simulations, they found that NHA9 demonstrated a non-random and non-core-shell spherical micelle-like organization at the oligomeric state. The experiments are well-designed and carefully validated, and their findings offered insights into the dynamic nature of NHA9 during condensate formation.

We thank the reviewer for this enthusiastic summary of our work!

Major points to address:

1. The schematic in Fig. 1a is confusing and does not show the removal of the GLEBS domain.

We thank the reviewer for pointing this out. In response, we have replotted Fig. 1a to illustrate the removal of the GLEBS domain (see revised Fig. 1a).

2. The GLEBS domain seems to be preceding the residues probed during the *in vitro* experiments. However, whether the overexpression assay in live cells and simulation uses full-length NHA9 or Δ GLEBS NHA9 is unclear. Would live cell expressing Δ GLEBS NHA9 show the same result as full-length NHA9? Similarly, would simulating Δ GLEBS NHA9 show the same result as full-length NHA9?

We thank the reviewer for the suggestion to better clarify the construct used in this study. As we used an NHA9 construct lacking the GLEBS domain for *in vitro* experiments, our simulations also applied the NHA9 construct lacking the GLEBS domain. We have added additional explanation in the main text. Furthermore, to assess whether simulations of NHA9- Δ GLEBS produce similar results to full-length NHA9, we performed a simulation of full-length NHA9, including the GLEBS domain for a nanocluster containing 10 chains. The mean density profiles of each block were similar for full-length NHA9 and NHA9 without GLEBS domain (see revised Supplementary Fig. 20). Furthermore, the mean end-to-end distances (R_E) and the apparent Flory exponents computed in both constructs were similar (see revised Supplementary Fig. 21).

In our cellular FLIM-FRET measurements, we used full-length NHA9 including the GLEBS domain to better mimic the physiological cellular environment. We have clarified this in the main text. We admit, that we initially shied away from doing the control experiments without GLEBS, as we did not expect any changes based on our understanding of that system, as those measurements are very time consuming. However, we have now tested whether cells expressing NHA9- Δ GLEBS exhibit the same assembly behavior as full-length NHA9. To this end, we measured three NHA9 mutants lacking the GLEBS domain using FLIM-FRET. All three mutants displayed similar inter-molecular distances to full-length NHA9 (see revised

supplementary Fig. 13), indicating that both full-length NHA9 and NHA9-ΔGLEBS adopt comparable assembly states in cells.

3. NHA9 was purified under denaturing conditions. Can the authors provide further validation that the protein refolds during *in vitro* experiments?

We thank the reviewer for the suggestion to provide additional validation of NHA9 refolding *in vitro*. In response, we conducted circular dichroism (CD) spectroscopy and protein thermal stability analyses (see revised supplementary Fig. 1).

As shown in Fig. 1b, disorder prediction indicated that only ~11% of residues in NHA9-ΔGLEBS are capable of folding. Thus, the corresponding folded signals in the CD spectra are expected to be weak. These “folded” residues are located within the HOXA9 domain and are predicted by AlphaFold to adopt helical conformations. To provide a comparison to a fully disordered NHA9 variant, we also purified an NHA9-ΔGLEBS-ΔHOXA9 without those 11% folded residues (FG domain), which is predicted to be fully disordered (see revised supplementary Fig. 1b). We then recorded CD spectra for both proteins (see revised supplementary Fig. 1c). In CD spectra, a negative band at 222 nm is characteristic of α -helical content, whereas disordered proteins typically exhibit a negative band near 200 nm. By comparison, NHA9-ΔGLEBS displayed a stronger signal at 222 nm ($[\theta]_{222} = -1630 \text{ deg}\cdot\text{cm}^2\cdot\text{dmol}^{-1}$) than the FG domain ($[\theta]_{222} = -1171 \text{ deg}\cdot\text{cm}^2\cdot\text{dmol}^{-1}$), whereas at 200 nm the FG domain exhibited a stronger signal ($[\theta]_{200} = -3059 \text{ deg}\cdot\text{cm}^2\cdot\text{dmol}^{-1}$) relative to NHA9-ΔGLEBS ($[\theta]_{200} = -2732 \text{ deg}\cdot\text{cm}^2\cdot\text{dmol}^{-1}$). These results suggest the presence of α -helical structures within the NHA9-ΔGLEBS sample.

In addition, because two tryptophan residues are located within the HOXA9 domain, we employed nano-differential scanning fluorimetry (nanoDSF), in which tryptophan fluorescence reports on protein unfolding. NHA9-ΔGLEBS samples were prepared in either native buffer or denaturing buffer containing 2 M guanidine hydrochloride, and subjected to a linear temperature ramp. Under native conditions, the melting temperature (T_m) was 48.8 °C, whereas in denaturing buffer it decreased to 42.7 °C (see revised supplementary Fig. 1d). The higher T_m under native conditions supports the existence of folded structures in the NHA9-ΔGLEBS sample.

Taken together, both CD spectroscopy and nanoDSF measurements demonstrate that NHA9-ΔGLEBS purified under denaturing conditions, can refold upon transfer into native buffer.

4. In the FCS data, it is unclear how the diffusion coefficient translate to cluster size. The authors made conclusions on nanocluster sizes but the data shown are all in diffusion coefficients (for example, Figs 2a, 3a).

We thank the reviewer for the suggestion to better explain how to convert diffusion coefficient to particle size. The Stokes-Einstein equation $D = \frac{k_B T}{6\pi\eta r}$ can be used to estimate the particle's hydrodynamic radius, where D is the translational diffusion coefficient, k_B is

Boltzmann's constant, T is the absolute temperature, η is the solvent viscosity, and r is the hydrodynamic radius of the particle. In response, we incorporated this equation into the methods section and clarified the description. However, because the Stokes-Einstein equation assumes spherical particles, the calculated radius may deviate from the true size of non-spherical particles. For this reason, we primarily report the diffusion coefficient for assessing particle size.

5. In addition, the 3-axis plot for FCS data is not helpful for visualizing the relative sizes of different peaks. May as well display them in 2-axis and separate to different concentrations similar to FRET data.

We thank the reviewer for pointing this out. To improve visualization, we replotted the diffusion coefficient distributions shown in Figures 2a and 3a, converting them from a two-dimensional peak display to a three-dimensional peak representation.

6. It will be preferred to include a negative control for Supp Fig. 1b showing NHA9 doesn't bind to a scrambled HOXA9 binding site.

We thank the reviewer for the suggestion to include a negative control in the electrophoretic mobility shift assay (EMSA). In response, we ordered a Cy5-labeled shuffled DNA fragment (5'-CTATATGCTATCGTGTAAACC-3', identical to the sequence applied in smFRET measurements shown in Supplementary Fig. 6) and repeated the EMSA using both the original DNA fragment (containing HOXA9 binding site) and the shuffled fragment containing a scrambled HOXA9 binding site (see revised Supplementary Fig. 2). A significant population of NHA9–DNA complexes was observed at 500 nM when incubated with the original DNA fragment, whereas no detectable complex formation was observed with the shuffled fragment. These results indicate that NHA9 binds strongly to the DNA fragment containing the HOXA9 binding site but shows weaker binding to the scrambled sequence, consistent with our smFRET measurements (see Supplementary Fig. 6). Furthermore, quantitative analysis of the gel band intensities allowed us to determine a dissociation constant (K_D) for NHA9 binding to the normal DNA fragment, yielding an apparent value of 0.3 μ M. Due to the heterogeneous nature of the NHA9 clusters, we would consider this an estimate.

7. The authors only measured the scaling component of NHA9 in the droplet phase in cell. How about the scaling component in the diffuse phase in cell?

We thank the reviewer for their comments regarding the measurement of scaling components of NHA9 in the dilute phase of cells. This is indeed an interesting and important topic. Our measurements of the much brighter droplets, in which the proteins are orders of magnitude more concentrated than in the soluble species, are already near the signal-to-noise limit of what we can achieve, due to cellular background fluorescence, background scattering, etc. In fact, we need to average across at least 20 cells to obtain a result that can be fitted with a unique solution. The NHA9 signal in the dilute phase is too weak to distinguish reliably from

the cellular background, as we noted in the discussion section of our current FLIM-FRET limitations: “Furthermore, smaller condensates with signals near background levels were excluded in our FLIM-FRET analyses, limiting our observations to larger structures.” With advancements in fluorescent dyes and more sensitive detectors, we look forward to being able to measure protein scaling components in the dilute phase within cells. However, the reality is that we are still very far away from being technically able to do this.

8. The authors showed representative images for labeling in cell (Figs. 5b,c). It is ideal to quantify the colocalizations across many different cells. Fig.5c can be compared to non-colocalization to heterochromatin marks such as H3K9me3.

We thank the reviewer for the suggestion. In response, we collected additional data on colocalisation across more cells. Specifically, we examined colocalisation between Genetic code expansion (GCE) labelled NHA9 and mEGFP–NHA9 fusion, as well as between GCE labelled NHA9 and H3K27ac (see revised supplementary Fig. 11). In all cases, the cells exhibited strong colocalisation.

We further assessed colocalization between GCE labelled NHA9 and H3K9me3. In this case, less GCE-labelled NHA9 showed colocalisation, in line with the trend observed in the literature for ectopic expression of NHA9 (Ahn, J. H. *et al.*, *Nature* **595**, 591–595, 2021).

Taken together, the consistent colocalisation with H3K27ac, compared to the weaker and variable colocalisation with H3K9me3, supports the successful labelling of NHA9 by our GCE system.

Minor points

1. Human proteins/genes should be upper case (e.g., NUP98).

We thank the reviewer. We have revised accordingly.

2. Line 65, “NHA9 condensates has” should be “have”.

We thank the reviewer. We have revised accordingly.

3. Line 231 should be Fig.2e not 4c. Also check and confirm other figure labelings are correct.

We thank the reviewer. We have revised accordingly.

4. Line 318, “...can be explain” should be “...can be explained”.

We thank the reviewer. We have revised accordingly.

5. Line 418 “blockcopolymer” should be two words.

We thank the reviewer. We have revised accordingly.

6. Fig. 6c: the authors concluded that the DNA-binding regions are mostly outside the cluster. But without showing the inside of the cluster, the readers are not convinced DNA-binding regions are not inside. It will be preferably to draw the cluster transparently to show the inside.

We thank the reviewer for this comment. As shown in Figure 6d, DBDs display a strong preference for localizing at the interface of NHA9 clusters. Nonetheless, we also observe a lower but significant concentration of DBDs within the cluster interior. We note that rendering the non-DBD beads transparent does not substantially improve visualization, as DBDs located on the opposite surface of the droplet become indistinguishable from those in the center. In Figure 6c, our goal was to highlight the enrichment of DBDs on the cluster surface. For the reviewer, we provide three alternative visualizations of a droplet with 50 chains in the figure below: one with fully opaque non-DBD beads, one with semi-transparent beads, and one with fully transparent beads.

Reviewer #4 (Remarks to the Author):

We thank the reviewer for the thorough evaluation of our work and for the constructive suggestions, which have helped us improve the quality of the manuscript.

Point-by-point

Reviewer comments in black

Our response in blue

Reviewer #1 (Remarks to the Author):

The authors have addressed the points that I have raised, and I am glad to see the quality of the paper have been improved by the work done by authors for the revision. Therefore, I recommend for publication.

Reviewer #1 (Remarks on code availability):

The repo contains simulation codes and analysis codes with README files including sufficient descriptions. I was able to download and visualize the example outputs.

We thank the reviewer for positive evaluation and thoughtful feedback throughout the review process. We are glad that the simulation and analysis codes were found to be clear and functional. We appreciate the reviewer's time and effort in evaluating our work and are pleased that revisions have strengthened the manuscript.

Reviewer #2 (Remarks to the Author):

I would like to sincerely thank the authors for taking the time to carefully respond to each of my points. The authors have done a commendable job, and I believe their revisions have strengthened the paper. I have no further comments or requests, and I look forward to seeing this exciting work published in Nature Communications.

We thank the reviewer for the positive feedback and encouraging remarks. We appreciate the reviewer's time and effort in evaluating our work and are pleased that revisions have strengthened the manuscript.

Reviewer #3 (Remarks to the Author):

The revised manuscript is much improved. I don't have further comments.

Reviewer #3 (Remarks on code availability):

N/A

We thank the reviewer for the positive feedback and encouraging remarks. We appreciate the reviewer's time and effort in evaluating our work and are pleased that revisions have strengthened the manuscript.

Reviewer #4 (Remarks to the Author):

We appreciate the reviewer's time and effort in evaluating our work and are pleased that revisions have strengthened the manuscript.